# Assessing biomedical knowledge robustness in large language models by query-efficient sampling attacks

**R. Patrick Xian**[1,◇]    **Alex J. Lee**[1]    **Satvik Lolla**[2]    **Vincent Wang**[1]

**Russell Ro**[2,1]    **Qiming Cui**[2,1]    **Reza Abbasi-Asl**[1,◇]

[1] *UC San Francisco*   [2] *UC Berkeley*

[◇] *xrpatrick@gmail.com, reza.abbasiasl@ucsf.edu*

**Reviewed on OpenReview:** *https://openreview.net/forum?id=pvol5JyVYB*

## Abstract

The increasing depth of parametric domain knowledge in large language models (LLMs) is fueling their rapid deployment in real-world applications. Understanding model vulnerabilities in high-stakes and knowledge-intensive tasks is essential to quantifying the trustworthiness of model predictions and regulating model use. The recent discovery of named entities as adversarial examples (i.e. adversarial entities) in natural language processing tasks raises questions about their potential impact on the knowledge robustness of pre-trained and finetuned LLMs in high-stakes and specialized domains. We examined the use of type-consistent entity substitution as a template for collecting adversarial entities for medium-sized billion-parameter LLMs with biomedical knowledge. To this end, we developed an embedding space, gradient-free attack based on powerscaled distance-weighted sampling for robustness evaluation, which has a low query budget and controllable coverage. Our method has favorable query efficiency and scaling over alternative approaches based on blackbox gradient-guided search, which we demonstrated for adversarial distractor generation in biomedical question answering. Subsequent failure mode analysis uncovered two regimes of adversarial entities on the attack surface with distinct characteristics. We also showed that entity substitution attacks can manipulate token-wise Shapley value explanations, which become deceptive in this setting. Our approach complements standard evaluations for high-capacity models and the results highlight the brittleness of domain knowledge in LLMs[1].

## 1 Introduction

LLMs such as pre-trained and finetuned generalist language models (GLMs) and their domain-adapted versions (Zhao et al., 2023; Wang et al., 2023a) incorporating transformer architectures are emerging as the technological backbone of next-generation search engines and knowledge bases (Petroni et al., 2019; Sung et al., 2021). They are increasingly used in knowledge-intensive tasks by the general public without sufficient assessment of inherent issues in trustworthiness and safety (Clusmann et al., 2023; Spitale et al., 2023; Si et al., 2023). Adversarial examples are valuable probes for model vulnerability and robustness, and are essential for performance improvements and regulatory audits (Shayegani et al., 2023). Adversarial attacks are classified by query scaling behavior into low- and high-query-budget attacks (see Fig. 1a). Collecting adversarial data at scale, either for adversarial training or robustness assessment, requires query-efficient attacks, first empirically studied in the vision domain (Ilyas et al., 2018; Cheng et al., 2019; Shukla et al., 2021). For language models, adversarial examples are generated through guided search (Alzantot et al., 2018), sampling and probabilistic methods (Ren et al., 2019; Yang et al., 2020; Yan et al., 2022), or collected dynamically by human operators (Jia & Liang, 2017; Wallace et al., 2019b). Most of these methods require a

---

[1] The code developed and datasets used for the work are available at https://github.com/RealPolitiX/qstab.

large amount of model evaluations (in the hundreds to thousands or beyond per instance) or whitebox access to model internals, and their evaluations tend to focus on small models (with < 1B parameters) (Maheshwary et al., 2021; Yu et al., 2024), which are insufficient in the era of LLMs. Sampling-based attacks obviate the need for predetermined search heuristics and can operate with only blackbox model access and a low query budget. Demonstrations of query-efficient attacks on LLMs meet an emerging demand for their continuous monitoring (Metaxa et al., 2021), which is particularly important for deploying LLMs in specialized domains.

Question answering (QA) has been the proving ground in recent achievements of biomedical language models (BLMs), here referring to GLMs adapted to the biomedical domain (Singhal et al., 2023; Kung et al., 2023; Liévin et al., 2024; Saab et al., 2024). Knowledge and reasoning components in biomedical QA assess the understanding of diverse concepts, which are represented by domain-specific named entities (NEs). The present work aims to identify adversarial entities in biomedical QA for both GLMs and BLMs. While GLMs may have acquired general biomedical knowledge during model development, BLMs gain further domain knowledge primarily through finetuning on a variety of text data from clinical notes (Lehman et al., 2023), electronic health records (Wornow et al., 2023), to abstracts or excerpts of scholarly publications (Tinn et al., 2023). Although QA is a frequently employed paradigm for model performance evaluation (Gardner et al., 2019; Robinson & Wingate, 2023), currently, biomedical QA still lacks domain-specific robustness assessments and its use in real-world applications, albeit promising, is still in its infancy (Kung et al., 2023). Evaluating model robustness in high-stakes, specialized domains requires customized perturbations apart from traditional benchmarks (Cecchini et al., 2024). Deployable LLMs in the biomedical domain should have robust contextual knowledge and language understanding of domain-specific entities.

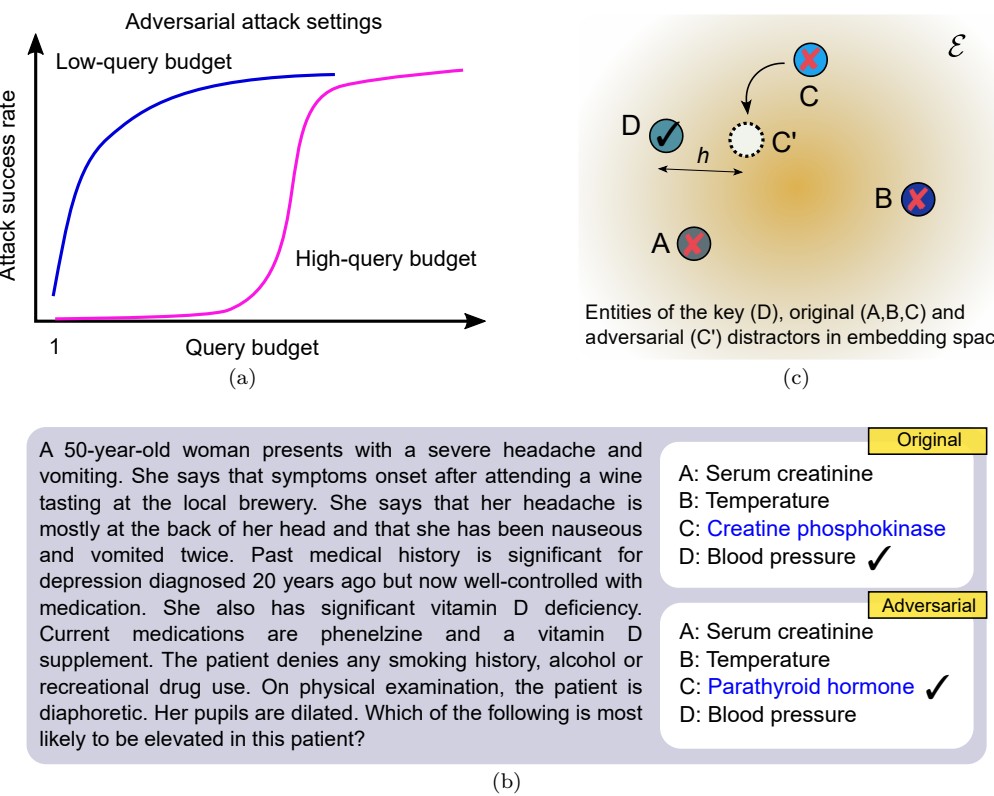

Figure 1: Entity substitution attack on QA with adversarial distractors. (a) Typical query scaling curves in low- and high-query-budget attack settings. (b) An adversarial distractor example found by type-consistent entity substitution (highlighted in blue). The correct (top) and incorrect (bottom) model responses (check-marked) before and after perturbation are included. (c) Illustration of the attack scheme in embedding space by PDWS for the example in (b). $\mathcal{E}$ represents the vocabulary set. D is the key to the question, A-C the original distractors, C′ is an adversarial distractor at distance $h$ from the key D.

In this work, we proposed and compared sampling- and search-based query-efficient adversarial attacks for evaluating the knowledge robustness of LLMs in the entity-rich, biomedical domain using type-consistent entity substitution (TCES). We investigated the setting where a large number of viable replacements ($N_{\mathcal{E}} > 1000$) per attack exist at only a small number of permissible text locations, which is prevalent in specialized domains where the NEs have shared characteristics such as their type and (word or phrase) structure. The setting differs from traditional synonym substitution (Liu et al., 2022; Yu et al., 2024), which have a significantly smaller number of viable replacements with matching word senses but can operate on many potential text locations. To relate to real-world scenarios (Apruzzese et al., 2023), we consider the attacks in the blackbox and hard-label setting with a limited query budget ($B \ll N_{\mathcal{E}}$). We used the attacks to generate adversarial examples of distractors (see Fig. 1b), or *adversarial distractors* in biomedical QA tasks.

Our main contributions include four aspects: (i) Introduction of powerscaled distance-weighted sampling (PDWS) method in embedding space to generate adversarial entities. Our work reports the first use of DWS for adversarial attacks in text[2]. The attack scheme has a broad coverage of the embedding space centered around an anchor point (illustrated as the key D in Fig. 1c) and maps the entity-level robustness landscapes of different LLMs in two regimes (near and far). (ii) Introduction of the `FDA-drugs` and `CTD-diseases` datasets as the source vocabularies of the perturbation sets, along with an entity type-annotated version of existing biomedical QA datasets MedQA-USMLE (Jin et al., 2021) and MedMCQA (Pal et al., 2022). Using these data, we demonstrated the vulnerability of LLMs to two of the most common NEs in the biomedical domain (Wei et al., 2019): drug and disease names (see example in Fig. 1b). (iii) We unified sampling- and gradient search-based methods in discrete text space and formulated TCES as an attack template, which we used for quantitative comparison of the scaling behaviors of the attacks in the same footing. We found that in the blackbox setting, sampling attacks can have favorable query scaling over gradient-based attacks, which are gradient quality-limited within a fixed query budget. (iv) By investigating the relationship between Shapley value-based model explanations (Chen et al., 2023) and adversarial attacks, we uncovered the defining signatures of successful attacks as a result of entity substitution.

## 2 Related works

**Text substitution attacks** Adversarial attacks in neural language models have been demonstrated from the character to the sentence level (Zhang et al., 2020). The majority of text substitution attacks employ typos (Gao et al., 2018) and synonyms (Mozes et al., 2021), which effectively impose lexical constraints. Word frequency has been used as a metric to select adversarial examples in (Mozes et al., 2021). Synonym substitution attacks have been demonstrated in sentiment analysis (Liu et al., 2022; Yu et al., 2024). Alzantot et al. (2018) used a genetic algorithm to search for adversarial word substitution. The sensitivity of models to entity-level perturbations, including substitution and swapping, creates problematic situations such as knowledge conflicts (Longpre et al., 2021) for knowledge-intensive tasks, compromised performance in machine reading comprehension (Yan et al., 2022) and table interpretation tasks (Koleva et al., 2023).

**Robustness in QA** The robustness of QA models has been studied using perturbation methods (Jia & Liang, 2017) and meaning-preserving transformations (Gan & Ng, 2019; Elazar et al., 2021). Sen & Saffari (2020) introduced a set of perturbations to assess language model generalization between QA datasets. Richardson & Sabharwal (2020) proposed a distractor design method using proximity in semantic space to probe model understanding of word senses. Awadalla et al. (2022) examined the connection between in- and out-of-distribution robustness for different LLMs. The performance of LLMs is sensitive to the ordering of choices (Pezeshkpour & Hruschka, 2023; Zheng et al., 2024). Overall, the brittleness of knowledge in LLMs remains a primary limitation for their applications (Elazar et al., 2021; Augenstein et al., 2024).

**Characteristics of adversarial examples** The defining characteristic of adversarial examples is the induction of false predictions in trained models that would otherwise predict accurately. Other characteristics include (i) their generality such as transferability and universality (Zou et al., 2023). Transferability refers to the effectiveness of the same example being adversarial to distinct models (Papernot et al., 2016). Universal

---

[2]Throughout the text, we use DWS to refer to the class of sampling method that employs distance-dependent probability weights. The powerscaled DWS is a type of DWS, so is the other version introduced in Wu et al. (2017).

adversarial perturbations are *reusable* across many examples tested on the same model (Moosavi-Dezfooli et al., 2017; Wallace et al., 2019a). (ii) The quality of adversarial examples may be judged by their perceptibility to human (Dyrmishi et al., 2023) or simple defense mechanisms such as grammar check. (iii) Adversarial attacks can operate in low- and high-query-budget settings (Shukla et al., 2021) with different query scaling behaviors. (iv) Adversarial examples with specifically designed perturbations can obstruct the model explanation (Noppel & Wressnegger, 2024) and render it unuseful.

## 3 Entity-centric adversarial attacks

### 3.1 Adversarial distractor generation

**Definition 3.1.** A multiple-choice question $\mathcal{Q}$ is represented by a tuple, $\mathcal{Q} = (\mathcal{C}, \mathcal{S}, \mathcal{D}, k)$, with the context $\mathcal{C}$, question stem $\mathcal{S}$, a distractor set $\mathcal{D} = \{d_i\}_{i=1}^l$, and a key $k$. The distractors are the wrong choices given to confuse test-takers, while the key is the correct one (CH & Saha, 2020).

**Definition 3.2** (Adversarial distractor). Given the original (or unperturbed) and a perturbed multiple-choice question pair, $\mathcal{Q} = (\mathcal{C}, \mathcal{S}, \mathcal{D}, k)$ and $\mathcal{Q}' = (\mathcal{C}, \mathcal{S}, \mathcal{D}', k)$, which differ by only a distractor, $\mathcal{D}' \setminus (\mathcal{D} \cap \mathcal{D}') = \{d'\}$. The distractor $d'$ is adversarial if the perturbed question elicits a false response from the model, while it answers correctly to the unperturbed question,

$$\text{argmax} \Pr_{\text{LLM}}(Y = k|\mathcal{C}, \mathcal{S}, \mathcal{D}', k) \neq \text{argmax} \Pr_{\text{LLM}}(Y = k|\mathcal{C}, \mathcal{S}, \mathcal{D}, k). \tag{1}$$

The threat model in Definition 3.2 refers to an untargeted attack, while the targeted version only requires restricting the answer to the perturbed question to a specific distractor. A simple way to construct the perturbed distractor set $\mathcal{D}'$ in $\mathcal{Q}'$ is to substitute a distractor $d$ by $d'$, or $\mathcal{D}' = (\mathcal{D} \setminus \{d\}) \cup \{d'\}$. The substitution is subject to the constraint $\|\mathcal{Q}' - \mathcal{Q}\| < \epsilon$, where $\epsilon$ is the perturbation budget, and the norm $\|\cdot\|$ refers to edit distance. For simplicity, we assume the distractors contain only NEs and only one NE each (such as in Fig. 1b). In Appendix A, we extend Eq. 1 to a more general case with multiple entities and additional non-entity text in the distractors.

**Example 3.1.** In the example of Fig. 1b, the stem $\mathcal{S}$ is the last sentence on the left side, "Which of the following is most likely to be elevated in this patient?" The context $\mathcal{C}$ is all the text before the stem, "A 50-year-old woman ... Her pupils are dilated." The original distractor set $\mathcal{D} = \{$Serum creatinine, Temperature, Creatine phosphokinase, Blood pressure$\}$, the perturbed distractor set $\mathcal{D}' = \{$Serum creatinine, Temperature, Parathyroid hormone, Blood pressure$\}$, the key $k$ is Blood pressure, or equivalently, D[3].

Unlike text addition attacks that introduce distracting information (Jia & Liang, 2017; Wallace et al., 2019a), the edit constraint on $\mathcal{Q}$ that is usually enforced in text adversarial attacks (Roth et al., 2024) doesn't preclude large semantic perturbations at the entity level in the distractors. In this work, we draw perturbations from a finite perturbation set, denoted as $\mathcal{E}_{-k} := \mathcal{E} \setminus \{k\}$, with $\mathcal{E} = \{e_j\}_{j=1}^{m+1}$ being the entity dataset as illustrated in Fig. 1a. In practice, $\mathcal{E}$ may be compiled from potential model use cases or curated by regulatory bodies in a model audit. Next, we discuss a unified view of discovering adversarial entities in the embedding space using blackbox (or gradient-free) attack methods based on sampling and search.

### 3.2 Powerscaled distance-weighted sampling

**Definition 3.3** (Text span representations). Given the token sequence representation of a span $e$, we define its corresponding vector representation in a text embedding of dimension $l$ as $\mathbf{e} \in \mathbb{R}^l$.

We consider sampling with distance-dependent probability weights from the embedding space of NEs. The concept of distance-weighted sampling (DWS) was previously introduced in contrastive learning in computer vision, where the Euclidean distance between samples and an anchor point was used and the probability weights depend on the embedding dimensionality (Wu et al., 2017). We propose a more flexible version

---

[3]For simplicity, in this work, we don't use different symbols to distinguish between the content of a distractor (or a key) from its index (e.g. A, B, C, D).

independent of the embedding dimensionality using a powerscaled distance with a tunable hyperparameter. Specifically, for a key $k$, we assign a distance-dependent probability weight $p_j \in [0, 1]$ to entity $e_j$ such that the distractor $d$ is substituted by

$$d' = e_j \quad \text{w.p.} \quad p_j = \frac{h^n(k, e_j)}{\sum_j h^n(k, e_j)}. \tag{2}$$

Here, $h(k, e_j)$ is a distance metric between the anchor point $k$ and $e_j$, while the exponent $n \in \mathbb{R}$ controls the shape of the probability weight distribution. When $n > 0$, as $n$ increases, the entities further away from $k$ are more preferably weighted. When $n < 0$, the sampling method employs inverse-distance-scaled weights, and the entities closer to $k$ are more preferably weighted. In the subsequent evaluations, we use $h(k, e_j) = \texttt{CosineDist}(\mathbf{k}, \mathbf{e}_j) = 1 - \mathbf{k} \cdot \mathbf{e}_j / (\|\mathbf{k}\| \cdot \|\mathbf{e}_j\|)$, calculated in a text embedding of choice, where $\mathbf{k}$ and $\mathbf{e}$ are the respective vector representations of $k$ and $e$. The formalism accommodates a deterministic regime as a limiting case[4], when $n \to +\infty$, and $p_j$ is vanishingly small for all but the entity furthest from $k$,

$$d' = e_j \quad \text{s.t.} \quad e_j = \operatorname*{argmax}_{e_j \in \mathcal{E}_{-k}} h(k, e_j). \tag{3}$$

When $n = 0$, $p_j$ is the same for each entity in $\mathcal{E}_{-k}$, corresponding to uniform random sampling. A short proof of the limiting cases is given in Appendix B. Therefore, our approach bridges the probabilistic and deterministic approaches to sampling adversarial examples.

### 3.3 Sampling view of zeroth-order adversarial attack

Blackbox attacks using approximate gradients may proceed with zeroth-order optimization (ZOO), where the gradients are estimated by finite difference to guide the search (Ilyas et al., 2018; Cheng et al., 2019). In the discrete text space, Berger et al. (2021) developed the DiscreteZOO attack, which contains three components: the word importance ranker, the candidate sampler, and the gradient-based optimizer. It operates at the word level and has been validated on BERT-sized models for synonym substitution attacks. The last two components aim to find a replacement text span $e'$ in each iteration through the gradient update rule

$$\mathbf{e}' = \mathbf{e}_0 - \lambda \widehat{\nabla}_{\text{ZO}} \text{LLM}(\mathcal{Q}; e_0). \tag{4}$$

Here, $e_0$ refers to the original text span, $\mathbf{e}'$ and $\mathbf{e}_0$ are the vector representations of $e'$ and $e_0$, respectively. $\lambda$ the learning rate, $\widehat{\nabla}_{\text{ZO}}$ indicates the zeroth-order gradient estimator, and $\widehat{\nabla}_{\text{ZO}}\text{LLM}(\mathcal{Q}; e_0)$ has the same dimensionality as $\mathbf{e}'$ or $\mathbf{e}_0$. We use $\text{LLM}(\mathcal{Q}; e_0)$ to denote the LLM text output with the unperturbed question $\mathcal{Q}$ (containing $e_0$) as the input, then the multi-point ($M \geq 2$) gradient estimate is constructed by querying the LLM $M$ times and computing

$$\widehat{\nabla}_{\text{ZO}}\text{LLM}(\mathcal{Q}; e_0) = \mathbb{E}_{\mathbf{u}} \left[ \frac{\Delta \text{LLM}(\mathcal{Q}; e_0)}{\Delta \mathbf{e}_0} \mathbf{u} \right] = \frac{1}{M} \sum_{i=1, e_i \in \mathcal{E}_{-k}}^{M} \frac{(\text{LLM}(\mathcal{Q}'; e_i) - \text{LLM}(\mathcal{Q}; e_0)) \cdot (\mathbf{e}_i - \mathbf{e}_0)}{\|\mathbf{e}_i - \mathbf{e}_0\|^2}. \tag{5}$$

This is an example of the random directions stochastic approximation (Nesterov & Spokoiny, 2017), where $\mathbf{u} = (\mathbf{e}_j - \mathbf{e}_0)/\|\mathbf{e}_j - \mathbf{e}_0\|$ is a randomly oriented vector in the embedding space corresponding to the neighboring entity $e_j$. Berger et al. (2021) computed Eq. 5 by random sampling of neighbors (in candidate sampler) within a similarity threshold for synonyms. The procedure was iterated once for all ranked attack locations. Given a noisy gradient estimate $\widehat{\nabla}_{\text{ZO}}\text{LLM}(\mathcal{Q}; e_0)$ and a discrete search space, we can rewrite Eq. 4 effectively as deterministic DWS in the embedding space by

$$d' = e_j \quad \text{s.t.} \quad e_j = \operatorname*{argmin}_{e_j \in \mathcal{E}_{-k}} h(e_0, e_j), \tag{6}$$

where $h(e_0, e_j) = \texttt{CosineDist}(\mathbf{e}_0 - \lambda \widehat{\nabla}_{\text{ZO}}\text{LLM}(\mathcal{Q}; e_0), \mathbf{e}_j)$, with $e_0$ as the anchor point, and argmin indicates the closest member within the text embedding. Using argmin is essential because the exact location $h(e_0, e_j)$

---

[4]The limit $n \to -\infty$ indicates replacing the distractor $d$ with $k$ at all times. Since $k$ is not in the perturbation set $\mathcal{E}_{-k}$, this limit is not attained.

in the discrete embedding space is usually not occupied. While a gradient estimate of sufficient quality is essential for a successful attack, increasing $M$ will count towards the query budget. Therefore, in the budgeted setting, a tradeoff should exist between the efficacy of the attack and the number of queries consumed for each gradient estimation.

## 4  Implementing attacks on biomedical QA

### 4.1  Dataset selection

**Entity datasets**  To relate to real-world scenarios, we sourced vocabulary datasets of drug and disease names from existing public databases. The drug names dataset (`FDA-drugs`) contains over 2.3k unique entities from known drugs approved by the United States Food and Drug Administration (FDA) and curated by Drug Central[5] (Ursu et al., 2017). FDA is a world authority that approves the legal distribution of drugs, therefore, the dataset represents the drug-entity names encountered in daily life. The disease names dataset[6] (`CTD-diseases`) contains over 9.8k unique entities from the Comparative Toxicogenomics Database (Davis et al., 2009), which contains a comprehensive collection of uniquely documented chemical-gene-disease interactions for humans. Both databases are curated by domain experts through regular updates, and unlike traditional text corpora (Mohan & Li, 2019), the entities in these two datasets contain no redundancy. The 2023 version of the datasets was used after minor data processing.

**Biomedical QA datasets**  We selected over 9.3k questions from the MedQA-USMLE (Jin et al., 2021) dataset and over 3.8k questions from the MedMCQA (Pal et al., 2022) dataset for benchmarking. Both datasets are entity-rich and cover a wide range of biomedical specialties and topics. The MedQA-USMLE dataset contains long-context questions that are used to assess medical students in the United States. The MedMCQA dataset contains short questions used in medical exams in India. They have been used in recent works to assess the biomedical knowledge in LLMs (Singhal et al., 2023; Han et al., 2023; Liévin et al., 2024; Saab et al., 2024). Both datasets are publicly available and don't contain personal information. We annotated the NEs according to the entity types of the Unified Medical Language System (UMLS) (Bodenreider, 2004) using `scispaCy` (Neumann et al., 2019). We then divided the QA datasets into drug- or disease-mention questions according to the UMLS entity types (see Appendix C) of NEs in distractors.

### 4.2  Type-consistent entity substitution (TCES)

Implementing attacks that target entities requires accounting for type consistency and compatibility with simultaneous multi-word (or span-level) operations, because of the inseparable nature of the entity components (e.g. growth hormone, acquired immunodeficiency syndrome). This effectively results in more substitution patterns (one to two words, two to one word, two to two words, two to three words, etc.) than the more common one-to-one synonym substitution (Mozes et al., 2021). We describe TCES in Algorithm 1 as a general and contextualized attack template for entities, with the goal of adversarially modifying model outputs with only one span-level substitution, specified by token boundaries. The template can accommodate single- or multi-query attacks and it applies to any task that may be formatted into QA (Gardner et al., 2019).

The initial steps in TCES involve entity recognition and typing (see Appendix C). For distractor generation, the type-filtered entities in the choices ($\text{Ent}_{\text{tfch}}$) are separated into key ($\text{Ent}_{\text{key}}$) and distractor ($\text{Ent}_{\text{distrc}}$) entities. In the biomedical domain, the entity type information is readily available from the UMLS (Bodenreider, 2004) semantic groups, such as *Chemicals & Drugs* and *Disorders* used for demonstration later. The `RankSelect` step selects the text span to attack ($\text{Ent}_{\text{victim}}$). In the PDWS attack, the type-matched entity in the distractors closest to the key is selected. The purpose is to maintain consistency and is not uniquely associated with the effects discussed later. In ZOO-based attacks, `RankSelect` corresponds to the word importance ranker. Then, the potential replacement entities are selected using `Sampler` (Eq. 2 for PDWS or Eq. 6 for DiscreteZOO), which also removes any duplicates of the key in the vocabulary ($\text{Ent}_{\text{vocab}}$). Sampling is carried out without replacement from the corresponding perturbation set $\mathcal{E}_{-k}$ (constructed with

---

[5]https://drugcentral.org/download

[6]https://ctdbase.org/downloads/

`FDA-drugs` or `CTD-diseases` datasets), equivalent to enforcing type consistency in the replacement entities with the entity to attack ($\text{Ent}_{\text{victim}}$). `GoalFunc` outputs a value from a loss function or from directly comparing with the answer to determine if the attack is successful.

### 4.3 Attack execution on LLMs

**Victim LLMs** We selected both generalist (GLMs) and specialist (BLMs) models for robustness evaluation via blackbox adversarial attacks. The criterion we adopted here for a model to have biomedical knowledge is that it should have a baseline performance better than random guessing (e.g. an accuracy of 0.25 for a specific four-choice QA task) in the evaluated domain-specific task. For GLMs, we used instruction-finetuned T5 (Flan-T5) series of models (Chung et al., 2024), and UL2 model (Tay et al., 2022) (Flan-UL2). For BLMs, we used MedAlpaca-7B (Han et al., 2023), MedLlama-3-8B-v2.0 from John Snow Labs (JSL-MedLM3-8Bv2)[7], Llama2-Medtuned-13B (LM2-Medtuned-13B) (Rohanian et al., 2024), and Palmyra-Med-20B (Kamble & Alshikh, 2023) from Writer. The selected open-source models have sizes in $\sim$ 1-20B range. Besides, we also evaluated GPT-3.5 from OpenAI.

**Attack settings** All models were evaluated at zero temperature or in the non-sampling setting and model inference was conducted in the zero-shot setting with only basic prompt instructions (see the prompt structure in Appendix D). Model inference of Palmyra-Med-20B (Kamble & Alshikh, 2023) used 4-bit quantization to improve speed (Dettmers & Zettlemoyer, 2023). The evaluation settings resulted in our somewhat lower baseline accuracies

---

**Algorithm 1** TCES attack template for collecting adversarial entities in distractors.

**Inputs**: Question Q, LLM, entity type $\tau$, budget $B$.
**Internals**: Number of choices $N_{\text{ch}}$, text t, entity Ent, embedding Emb, key or correct answer k.
**Outputs**: Number of queries and the replacement entity, None if unsuccessful after all attempts.

$\text{TCESATTACKER}(Q, \text{Model}, \tau, B)$
  $\text{t}_{\text{choices}}, \text{k} \leftarrow \text{Q.choices}, \text{Q.answer}$
  **for** $c \leftarrow 1$ to $N_{\text{ch}}$ **do**
   $\text{t}_{\text{ch}} \leftarrow \text{t}_{\text{choices}}[c]$
   $\text{Ent}_{\text{ch}} \leftarrow \text{NERecognizer}(\text{t}_{\text{ch}})$
   $\text{Ent}_{\text{tfch}} \leftarrow \text{TypeFilter}(\text{Ent}_{\text{ch}}, \tau)$
  $\text{Ent}_{\text{key}}, \text{Ent}_{\text{distrc}} \leftarrow \text{SplitByLabel}(\text{Ent}_{\text{tfch}})$
  $\text{Ent}_{\text{victim}} \leftarrow \text{RankSelect}(\text{Ent}_{\text{key}}, \text{Ent}_{\text{distrc}}, \text{Emb})$
  $i \leftarrow 0$
  **while** $i < B$ **do**
   $\text{Ent}_{\text{perturb}} \leftarrow \text{Sampler}(\text{Ent}_{\text{key}}, \text{Ent}_{\text{victim}},$
         $\text{Ent}_{\text{vocab}}, \text{Emb})$
   $Q' \leftarrow Q (\text{Ent}_{\text{victim}} \rightarrow \text{Ent}_{\text{perturb}})$
   $\text{k}' \leftarrow \text{Model}(Q')$
   **if** $\text{GoalFunc}(\text{k}', \text{k}) \leftarrow 1$ **then**
    **return** $i, \text{Ent}_{\text{perturb}}$
   $\text{Ent}_{\text{vocab}} \leftarrow \text{Ent}_{\text{vocab}} \setminus \text{Ent}_{\text{perturb}}$
   $i \leftarrow i + 1$
  **return**

---

than the reported ones which are often achieved with the help of few-shot prompting and prompt optimization. The DiscreteZOO attack (Berger et al., 2021) was originally implemented in the TEXTATTACK framework (Morris et al., 2020b), which we updated to be compatible with span-level perturbation using its boundaries, multi-GPU split-model inference, and decoder-based billion-parameter LLMs by modifying the attack recipe. The following three aspects of the benchmarks are in our focus.

- **Query budget**: We used fixed query budgets ($B$) for three main types of attacks: (i) Single-query sampling-based attacks were used as the reference because the DiscreteZOO attack requires a minimum of 3 model queries; (ii) Multi-query attacks used a budget of 8 for reasonable computational cost across all models and attack settings for both sampling- and search-based attacks; (iii) The query scaling trends of specific LLMs and different attack settings were investigated with a series of query budgets under 100 per input instance. Here, we also included the deterministic versions of PDWS to sample the nearest and farthest elements to an anchor point. The element was taken from the perturbation set and the distances (between entities) were calculated using cosine in a chosen text embedding. Their query-limited versions are referred to as $B$-nearest and $B$-farthest element sampling, respectively, where all $B$ nearest or farthest elements were used sequentially to attack models in query scaling studies. Detailed definitions are given in Appendix B.4.

---

[7]https://huggingface.co/johnsnowlabs/JSL-MedLlama-3-8B-v2.0

- **Text embedding**: Both the PDWS and DiscreteZOO attacks require sampling from a text embedding. The counter-fitted GloVe embedding (Mrkšić et al., 2016) was the default for DiscreteZOO, but GloVe is poor for representing biomedical vocabulary (Wang et al., 2018). We chose two recent embedding models with improved latent representations of text relations for the benchmarks: (i) CODER (Yuan et al., 2022), a BERT-structured embedding model contrastively pretrained to distinguish UMLS knowledge graph concepts; (ii) GTE-base (Li et al., 2023b) from contrastive learning of general web-scale text data[8].

- **Core hyperparameters**: For single-query PDWS attacks, we tuned the hyperparameter $n$ within the interval of [-50, 50] using grid search with a step of 5 or 10 because the local maxima of ASR appear on the positive and negative sides. For multi-query PDWS attacks, hyperparameter $n$ was briefly re-tuned around its optimal value in the single-query attack. Most tuned hyperparameters fall within [-30, -5] and [5, 30]. Operating the DiscreteZOO attack in a compatible setting for entities requires two key modifications: (i) Use a random word ranking, which allows the attack algorithm to operate in the low-query-budget setting. (ii) Disable the similarity threshold for candidate point selection in gradient estimation because only entity type consistency is required. The ASR was greatly improved after this step. Benchmarks of the DiscreteZOO attacks were carried out with these settings.

## 5 Finding and assessing adversarial entities

### 5.1 Two regimes of adversarial entities

**Diversity quantification of adversarial entities**    The embedding space picture of the substitution attack in Fig. 1a provides a geometric interpretation of PDWS to examine different types of adversarial examples by tuning $n$. We use the Gini-Simpson index (Rao, 1982), $\eta_{\text{GS}} = 1 - \sum_j q_j^2$, to quantify the diversity of adversarial examples and a low value indicates less diversity. Here, $q_j \in (0, 1]$ is the probability of occurrence of the entity $e_j$ among all instances of adversarial entities obtained from sampling, and $\sum_j q_j = 1$. A low diversity index indicates high reusability and is a proxy for the universal adversarial attack regime (Moosavi-Dezfooli et al., 2017; Wallace et al., 2019a).

Considering the joint effects of diversity and hyperparameter $n$ on ASR in Fig. 2 reveals two regimes: (i) adversarial entities semantically close to ($n < 0$) the key $k$ have a greater diversity; (ii) adversarial entities semantically far from ($n > 0$) the key $k$ are more reusable. TCES in either of these two regimes can lead to an increase in ASR from random sampling (equivalent to $n = 0$), as shown in Tables 1-2. Fig. 2 shows that a larger positive $n$ in Eq. 2 samples more universal adversarial entities than in other situations, leading to a marked drop in the diversity index. Moreover, there exist local maxima of ASR in both regimes with finite $n$, which were discovered by hyperparameter tuning. We call the behavior the *two-regime effect* and it was observed in almost all models evaluated here in varying magnitudes for both datasets, thanks to the controllable spatial coverage of sampling methods incorporating distance information. In both single- and multi-query attacks, the benchmarks show that PDWS attacks in the $n < 0$ regime tend to incur a greater change in model performance than in the $n > 0$ regime, yet prominent counterexamples exist, including MedAlpaca-7B and Llama2-Medtuned-13B evaluated on drug-mention questions, and Flan-T5-xl evaluated on disease-mention questions in Table 1.

**Semantic distortion by adversarial entities**    Besides ASR, evaluating the semantic distortion is another way to quantify the effectiveness of adversarial attacks. We estimated the semantic distortion of entire prompts in successful attacks before and after entity substitution, corresponding to $\mathcal{Q}$ and $\mathcal{Q}'$ in Definition 3.2. We calculated the average prompt semantic similarity (PSS) using `SentenceTransformer` (Reimers & Gurevych, 2019) with the RoBERTa-large model (`all-roberta-large-v1`)[9]. At each distinct $n$, the PSS $\in [-1, 1]$ with 1 being the most similar, is averaged over all successful attack instances to obtain the average PSS, a quantitative measure of semantic distortion. Its dependence on the hyperparameter $n$ for different models and datasets is shown in Fig. 2.

---

[8]CODER at https://huggingface.co/GanjinZero/UMLSBert_ENG; GTE-base at https://huggingface.co/thenlper/gte-base.
[9]https://huggingface.co/sentence-transformers/all-roberta-large-v1

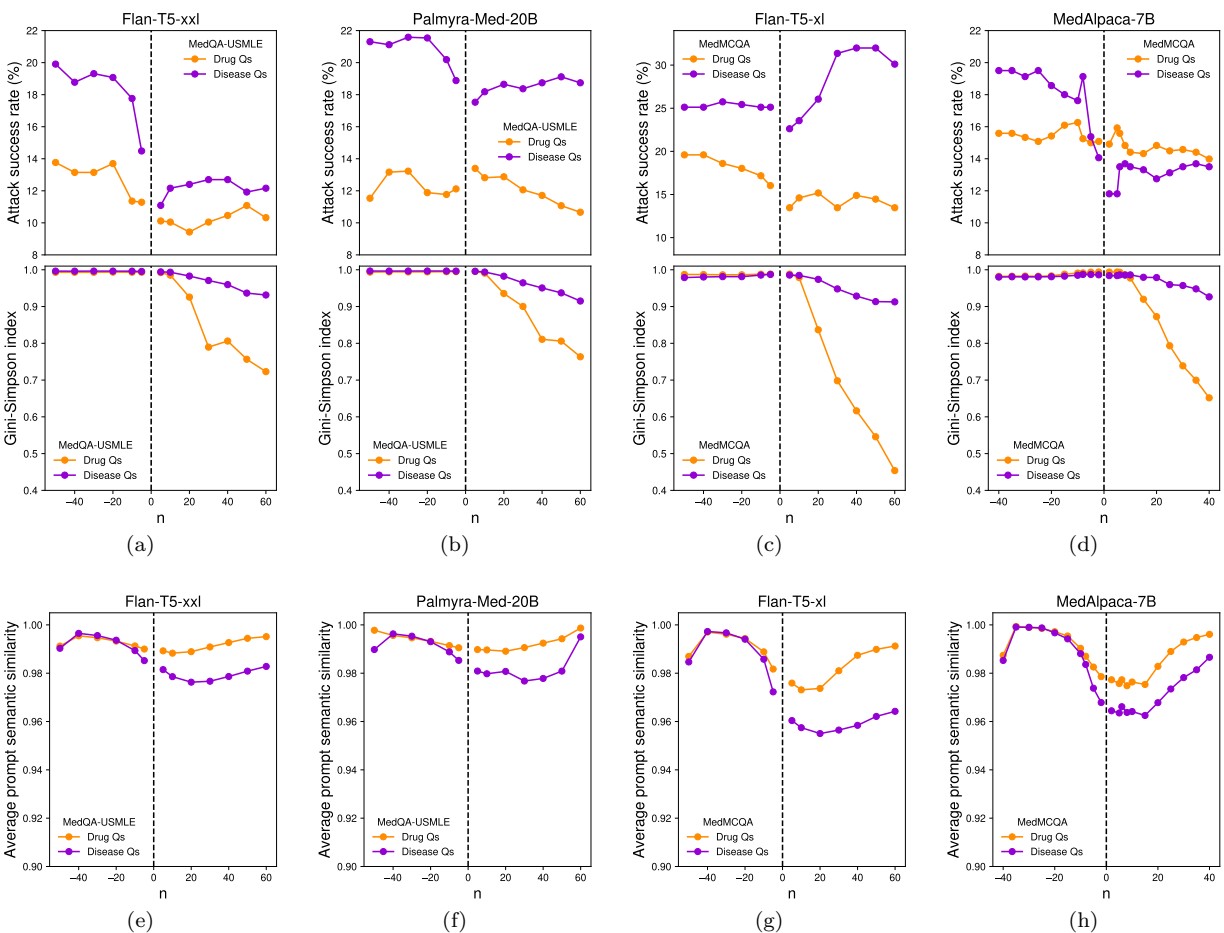

Figure 2: Powerscaled DWS of adversarial distractors exhibits a two-regime effect at negative and positive $n$ values (see Eq. 2) in ASR (top) and diversity index (bottom) of replacement entities in successful attacks. Local maxima in ASR with a finite $n$ are also present in each regime. The vertical dashed line indicates the location of random sampling. The observed similar behaviors are compared across models and datasets in (a) Flan-T5-xxl on MedQA-USMLE, (b) Palmyra-Med-20B on MedQA-USMLE, (c) Flan-T5-xl on MedMCQA, (d) MedAlpaca-7B on MedMCQA. Disease and drug-mention questions are separated by colors. The average prompt semantic similarity displayed in (e)-(h) is calculated for the successful attacks obtained from the corresponding attack settings in (a)-(d), respectively.

**Adversarial entity characteristics** The two-regime effect indicates that the success of the attacks may come from two sources that dominate at different locations on the same attack surface, which refers to entity substitution in the present work.

- At larger distances, the attack succeeds potentially due to the obscurity of the distractor entities in the question context, which can also profoundly impact the model performance (Li et al., 2023a). This explanation may be further supported by examining the diversity index, $\eta_{\mathrm{GS}}$, which shows the dominance of a few replacement entities in successful attacks at large distances. We found these highly reusable adversarial entities by ranking their occurrences in successful attacks by powerscaled DWS. For drug names, the top-ranked adversarial entities include n-acetylglucosamine (an amino sugar and anti-inflammatory drug) and technetium Tc 99m exametazime (a radiopharmaceutical and contrast agent)[10]. For disease names, some of the chromosome deletion syndromes and rare disease names are more reusable than others. These

---

[10]The RxList drug index contains entries for n-acetylglucosamine and technetium Tc 99m exametazime.

| Model (size) | Entity mention | MedQA-USMLE | | | MedMCQA | | |
|---|---|---|---|---|---|---|---|
| | | Baseline (Acc) | RandS@1 (ASR %) | PDWS@1 (-/+) +CODER (ASR %) | Baseline (Acc) | RandS@1 (ASR %) | PDWS@1 (-/+) +CODER (ASR %) |
| Flan-T5-xl (3B) | Drugs | 0.318 | 10.4 | **13.5** / 10.7 | 0.265 | 14.3 | **19.6** / 15.1 |
| | Diseases | 0.335 | 11.0 | **16.1** / 12.5 | 0.261 | 23.8 | 25.7 / **31.8** |
| Flan-T5-xxl (11B) | Drugs | 0.323 | 9.6 | **13.6** / 11.2 | 0.291 | 12.7 | **19.6** / 16.2 |
| | Diseases | 0.345 | 12.2 | **20.0** / 12.8 | 0.376 | 15.2 | **26.6** / 23.9 |
| Flan-UL2 (20B) | Drugs | 0.382 | 8.6 | **12.3** / 10.5 | 0.416 | 13.9 | **15.1** / 12.7 |
| | Diseases | 0.396 | 10.1 | **15.7** / 13.1 | 0.408 | 15.7 | **25.5** / 17.7 |
| GPT-3.5 (175B)* | Drugs | 0.438 | 9.1 | **11.2** / 10.7 | | | |
| | Diseases | 0.453 | 8.6 | **11.5** / 8.8 | | | |
| MedAlpaca-7B | Drugs | 0.479 | 7.2 | 8.2 / **9.6** | 0.448 | 13.9 | **16.2** / 15.9 |
| | Diseases | 0.498 | 8.4 | **12.5** / 9.2 | 0.434 | 14.8 | **19.1** / 13.7 |
| JSL-MedLM3-8Bv2 | Drugs | 0.569 | 4.9 | **8.4** / 6.1 | 0.827 | 10.4 | 10.2 / **11.8** |
| | Diseases | 0.592 | 6.5 | **11.4** / 6.4 | 0.771 | 9.5 | **18.5** / 11.3 |
| LM2-Medtuned-13B | Drugs | 0.371 | 16.4 | **18.1** / 17.8 | 0.568 | 15.3 | 17.3 / **18.1** |
| | Diseases | 0.398 | 19.6 | **23.1** / 19.6 | 0.515 | 19.0 | **27.6** / 21.0 |
| Palmyra-Med-20B | Drugs | 0.382 | 12.6 | 13.3 / **13.5** | 0.614 | 10.2 | **13.8** / 12.7 |
| | Diseases | 0.441 | 16.9 | **21.5** / 18.7 | 0.624 | 15.4 | **20.1** / 11.9 |

Table 1: Single-query (budget = 1) attack success rates (ASRs) on subsets of MedQA-USMLE and MedM-CQA datasets by perturbing the drug or disease entity mentions. Zero-shot baseline (Baseline) accuracy is included for reference next to the ASR of sampling attacks using perturbation by random sampling (RandS), powerscaled distance-weighted sampling (PDWS). The "-" and "+" signs indicate the two regimes ($n < 0$ and $n > 0$). For each model, the highest ASR for a type of entity-mention question in each dataset is bolded. The model with an asterisk ($*$) were only evaluated on one dataset to limit the computational cost.

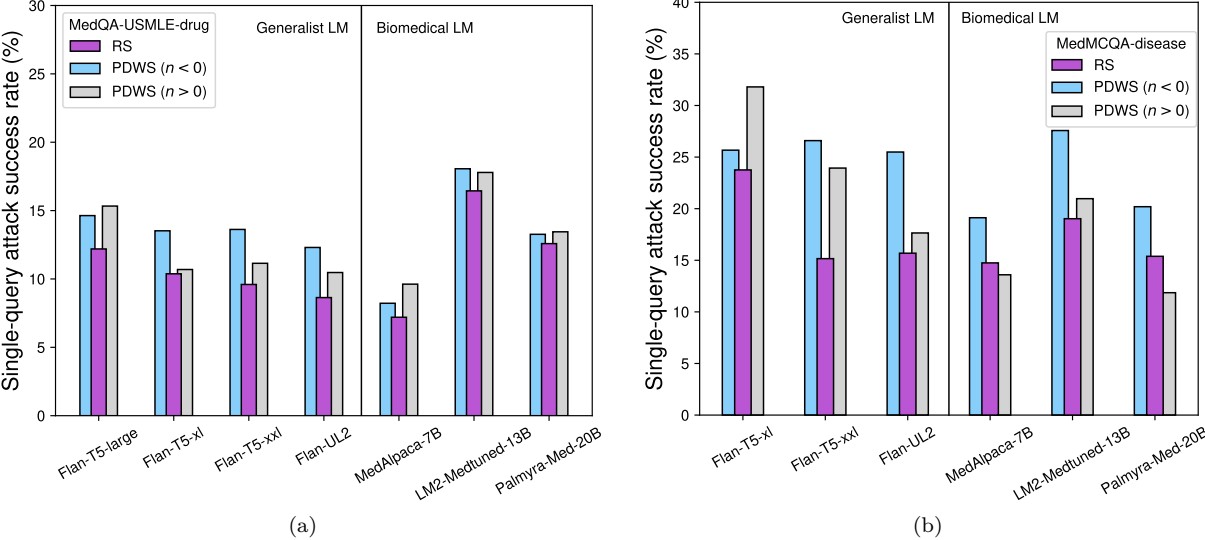

(a)  (b)

Figure 3: Model robustness from single-query adversarial attacks using results in Table 1. The models (GLMs and BLMs) were evaluated on drug-mention questions from MedQA-USMLE (left) and disease-mention questions from MedMCQA (right) datasets. The GLMs and BLMs are ordered horizontally by their sizes. The bar colors distinguish between different attack methods. Perturbations by random sampling (RS) are in purple. Perturbations by powerscaled DWS (PDWS) in the $n < 0$ and $n > 0$ regimes are in blue and grey, respectively.

entities dominate the sampling results in the $n > 0$ regime for all LLMs, indicating their transferability and universality.

- At small distances, semantic proximity (Mozes et al., 2021) reduces the distinguishability between the key and the distractor, thereby increasing the chance of model failure. Examples include similar drugs or diseases with insufficient details such as using diabetes as a distractor instead of the correct answer diabetes II (refers to type-II diabetes) can result in a successful attack.

- Adversarial distractors result in little semantic distortion at the prompt level (see Fig. 2e-h). The experiments on the MedMCQA dataset yield somewhat lower average PSS than MedQA-USMLE because of the longer format of the latter dataset, leading to slightly larger distortions by entity substitution. Adversarial entities obtained in the $n < 0$ regime results in higher average PSS compared to those in the $n > 0$ regime, while there is also a two-regime effect – the average PSS tends to be the lowest around $n = 0$ and higher when $n$ departs further from 0. The trend is consistent for different models and datasets demonstrated in Fig. 2.

| Model (size) | Entity mention | MedQA-USMLE (ASR %) | | | | MedMCQA (ASR %) | | | |
|---|---|---|---|---|---|---|---|---|---|
| | | RandS @8 | PDWS@8 (-/+) +CODER | DZOO@8 +CODER | DZOO@8 +GTE-base | RandS @8 | PDWS@8 (-/+) +CODER | DZOO@8 +CODER | DZOO@8 +GTE-base |
| Flan-T5-xl (3B) | Drugs | 22.0 | **29.3** / 23.3 | 21.7 | 19.5 | 32.5 | **48.3** / 36.2 | 34.3 | 28.2 |
| | Diseases | 27.2 | **38.8** / 24.8 | 20.9 | 23.2 | 47.1 | **61.7** / 49.0 | 45.8 | 39.3 |
| Flan-UL2 (20B) | Drugs | 22.5 | **27.5** / 22.3 | 20.4 | 19.2 | 26.4 | **35.6** / 22.4 | 30.7 | 26.4 |
| | Diseases | 25.8 | **34.9** / 26.8 | 25.7 | 24.0 | 39.2 | **49.5** / 31.6 | 41.3 | 37.0 |
| MedAlpaca-7B | Drugs | 16.8 | 21.1 / 22.3 | **22.8** | 20.5 | 27.9 | 37.1 / 30.1 | **42.6** | 36.8 |
| | Diseases | 21.7 | **30.3** / 24.0 | 26.5 | 25.2 | 29.8 | **43.9** / 30.8 | 40.5 | 38.1 |
| Palmyra-Med-20B | Drugs | 30.1 | **31.4** / 31.3 | 29.1 | 29.6 | 28.5 | **37.3** / 31.9 | 35.4 | 35.0 |
| | Diseases | 33.2 | **35.1** / 30.2 | 32.3 | 34.5 | 41.1 | 48.4 / 39.6 | 46.3 | **50.1** |

Table 2: Multi-query (here budget = 8) attack success rates (ASR %) on various LLMs by perturbing the drug and disease entity mentions in MedQA-USMLE and MedMCQA datasets. The PDWS attack uses the embedding from CODER and DiscreteZOO (DZOO) attacks use either CODER or GTE-base embedding. For each model, the highest ASR for a type of entity-mention question in each dataset is bolded.

## 5.2 Scaling characteristics from adversarial entities

**Size is not all for scaling model robustness.** We compare model robustness using single-query ASR as the metric in Table 1 and Fig. 3. For the Flan-T5 series of GLMs (Chung et al., 2024) and Flan-UL2 (Tay et al., 2022), the single-query ASR is more pronounced in smaller and less performant models, indicating the improvement of robustness as the model's size grows. However, an opposite trend of robustness is seen in BLMs. It is worth noting that although Palmyra-Med-20B (Kamble & Alshikh, 2023) has about three times the number of parameters than MedAlpaca-7B (Han et al., 2023), the larger model is noticeably more sensitive to entity perturbation. In terms of ASR, the BLMs are comparable to the GLMs in each type of questions evaluated on. Another observation is that the ASR for models is asymmetrical under PDWS attacks. For most of the models, sampling adversarial entities with inverse-distance-scaled weights ($n < 0$ in Eq. 2) induces more performance drop than sampling with distance-scaled weights ($n > 0$ in Eq. 2), yet prominent counterexamples exist. The attack performance is further investigated in the sequential, multi-query setting (budget = 8) in Table 2, where we also compared sampling- against search-based DiscreteZOO attacks. Here, PDWS attacks in the $n < 0$ regime are still largely the most effective.

**Sampling attacks has improved query scaling over blackbox gradient attacks in specialized text embedding.** A query scaling study of adversarial attacks investigates attack success with varying query budgets per input instance (Shukla et al., 2021). It is a comprehensive evaluation of attack methods. Fig. 4 compares the query scaling of both sampling- and search-based methods for TCES. For low-query-budget attacks (Fig. 1a), each scaling curve exhibits a rapid rise phase and a plateau phase, separated by an inflection point. The key message here is that the advantage of PDWS attacks over DiscreteZOO depends on the model and the choice of text embedding. The PDWS attacks are overarchingly more effective for GLMs than DiscreteZOO, while for BLMs, the advantage is reduced. Moreover, using a specialized text

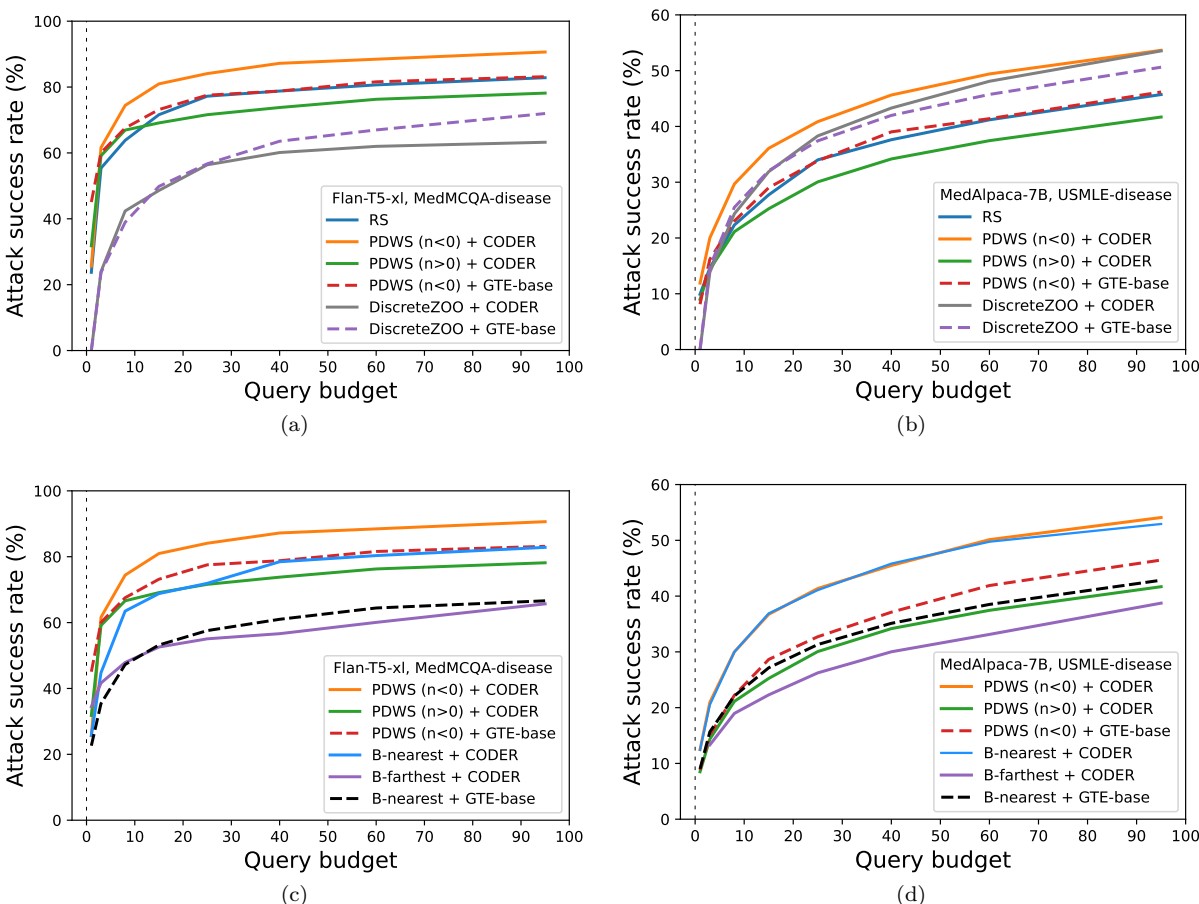

Figure 4: Example scaling curves of the query budget against ASR for disease-mention questions with (a) Flan-T5-xl model on MedMCQA dataset and (b) MedAlpaca model on MedQA-USMLE dataset. The curves are generated using TCES of disease names with the methods described in the legends. The DiscreteZOO attacks were run in the low-query-budget setting. Executing the random sampling (RS) attack doesn't require a text embedding, while the other attacks were evaluated with the CODER or GTE-base as text embedding. In (c) and (d), query scaling of the attacks based on $B$-nearest and $B$-farthest element sampling ($B$ is query budget) for the same datasets and models as in (a) and (b) are compared with PDWS.

embedding further boosts the advantage of PDWS attacks compared with a general text embedding due to the difference in embedding space neighborhood. We expand the analysis in the following along three directions.

- **Query scaling depends on text embedding**: Changing the text embedding affects the query scaling behavior of both PDWS and DiscreteZOO attacks. Fig. 4 shows that the influence of the embedding choice on PDWS is more pronounced in the rapid rise phase, while it influences the DiscreteZOO primarily in the plateau phase. Specifically, changing the text embedding from domain-specialized CODER (Yuan et al., 2022) to the general GTE-base (Li et al., 2023b), the performance advantage of PDWS over DiscreteZOO attacks shrunk in Flan-T5-xl (Fig. 4a) and inverted in MedAlpaca-7B (Fig. 4b).

- **Sampling hyperparameter controls scaling curve shape in PDWS attacks**: The hyperparameter $n$ changes the inflection point of the scaling curve. When the query budget passes the inflection point of the curve, random sampling starts to surpass PDWS ($n > 0$) in ASR. This is a consequence of the distribution of adversarial entities in the embedding space and the regional bias derived from the sampling weights. The examples in Fig. 4 show that the largest performance gap between the PDWS and DiscreteZOO attacks appears around the inflection point of the respective query scaling curve.

- **Gradient quality affects query scaling in DiscreteZOO attacks**: The query complexity in blackbox gradient estimation is a long-standing theoretical topic in derivative-free optimization (L'Ecuyer & Yin, 1998) and online learning (Duchi et al., 2015). Nevertheless, how gradient quality affects query scaling remains less studied for adversarial attacks employing gradient-based discrete optimization. Gradient estimation consumes a large part of the total query budget, leading to a tradeoff between the gradient quality and terminal ASR attainable (at max budget) in the query scaling curve. We empirically explored the query scaling behavior of the DiscreteZOO attack in Appendix E with different candidate points in the multi-point gradient estimation, which in effect alters the gradient quality. The results show that having too few candidate points (i.e. small $M$ in Eq. 5) leads to an early onset of the plateau phase, thereby limiting the highest attainable ASR, whereas having many candidate points can lead to a slower growth of ASR in the rapid rise phase of the query scaling curve.

---

A 73-year-old man presents to the outpatient clinic complaining of chest pain with exertion. He states that resting for a few minutes usually resolves the chest pain. Currently, he takes 81 mg of aspirin daily. He has a blood pressure of 127/85 mm Hg and heart rate of 75/min. Physical examination reveals regular heart sounds and clear lung sounds bilateral. Which medication regimen below should be added?

| | |
|---|---|
| A: Amlodipine daily. Sublingual nitroglycerin as needed.
B: Metoprolol and a statin daily. Sublingual nitroglycerin as needed. ✓
C: Metoprolol and ranolazine daily. Sublingual nitroglycerin as needed.
D: Amlodipine and a statin daily. Sublingual nitroglycerin as needed. | A: Amlodipine daily. Sublingual nitroglycerin as needed.
B: Metoprolol and a statin daily. Sublingual nitroglycerin as needed.
C: N-acetylglucosamine and ranolazine daily. Sublingual nitroglycerin as needed.
D: Amlodipine and a statin daily. Sublingual nitroglycerin as needed. ✓ |

---

[Context]: A 73-year-old man presents to the outpatient clinic complaining of chest pain with exertion. He states that resting for a few minutes usually resolves the chest pain. Currently, he takes 81 mg of aspirin daily. He has a blood pressure of 127/85 mm Hg and heart rate of 75/min. Physical examination reveals regular heart sounds and clear lung sounds bilateral.
[Question]: Which medication regimen below should be added?
A: Amlodipine daily. Sublingual nitroglycerin as needed.
B: Metoprolol and a statin daily. Sublingual nitroglycerin as needed.
C: Metoprolol and ranolazine daily. Sublingual nitroglycerin as needed.
D: Amlodipine and a statin daily. Sublingual nitroglycerin as needed.

Unperturbed

[Context]: A 73-year-old man presents to the outpatient clinic complaining of chest pain with exertion. He states that resting for a few minutes usually resolves the chest pain. Currently, he takes 81 mg of aspirin daily. He has a blood pressure of 127/85 mm Hg and heart rate of 75/min. Physical examination reveals regular heart sounds and clear lung sounds bilateral.
[Question]: Which medication regimen below should be added?
A: Amlodipine daily. Sublingual nitroglycerin as needed.
B: Metoprolol and a statin daily. Sublingual nitroglycerin as needed.
C: N-acetylglucosamine and ranolazine daily. Sublingual nitroglycerin as needed.
D: Amlodipine and a statin daily. Sublingual nitroglycerin as needed.

Adversarially perturbed

Figure 5: (Top) An entity substitution attack using n-acetylglucosamine as the replacement entity for metoprolol creates an adversarial distractor. (Bottom) Heatmaps of token-wise Shapley values for a question before and after the adversarial attack on choice C. The model prediction changes from the correct choice of B (unperturbed) to the incorrect choice of D (adversarially perturbed).

- **Deterministic sampling attacks are competitive proxies of their probabilistic counterparts.** Performance comparison between $B$-nearest and $B$-farthest element sampling, the deterministic and query-limited versions of PDWS (corresponding to the $n < 0$ and $n > 0$ regimes) are given in Fig. 4c-d for the two example cases studied for the query scaling. The results show that the deterministic sampling attacks are already competitive against DiscreteZOO and also close in performance to PDWS

in both specialized and general text embeddings. The difference is that the two types of deterministic sampling attacks examine only predefined regions in the embedding space and cannot explore the entire attack surface as PDWS would through tuning the $n$ hyperparameter.

### 5.3 Adversarial entities manipulate explainability

Post-hoc analysis is a common way to understand the rationale behind model decisions (Murdoch et al., 2019). Score-based feature importance such as that based on the Shapley values is routinely used to construct post-hoc explanation of model prediction through feature attribution (Chen et al., 2023). The stability of model explanation is a growing concern for their proper use (Alvarez-Melis & Jaakkola, 2018; Hancox-Li, 2020; Lin et al., 2023). Investigations on the susceptibility of LLM explanations to adversarial perturbation in QA are still lacking. For score-based feature importance, an explainer $\phi(\cdot)$ maps the $m$-dimensional feature $\mathbf{x} \in \mathbb{R}^m$ to scores $\phi(\mathbf{x}) \in \mathbb{R}^m$. In the current context, we assess the changes in feature importance represented by Shapley values (Chen et al., 2023) before ($\phi(\mathcal{Q})$) and after ($\phi(\mathcal{Q}')$) the adversarial attack by entity substitution $\mathcal{Q} \to \mathcal{Q}'$. For convenience of discussion, we use *Shapley profile* to refer to $\phi(\mathcal{Q})$ (or $\phi(\mathcal{Q}')$), the token-wise Shapley values calculated over $\mathcal{Q}$ (or $\mathcal{Q}'$).

We investigated the changes in the Shapley profile in the two regimes of adversarial entities for open-source LLMs, because the calculation requires access to model weights. We found that the token with the largest Shapley value (top-1 feature) is highly correlated with the model's prediction before and after adversarial perturbation, with an example shown in Fig. 5 produced using Flan-T5-large. This behavior agrees with the symbol-binding characteristic of LLMs in multiple-choice QA (Robinson & Wingate, 2023). Moreover, the Shapley profile typically exhibits pronounced changes before and after entity perturbation, especially in the text locations close to the perturbed entities (Fig. 5 and Appendix E). Together, these observations indicate that the substitution attacks effectively manipulate model explanations, because the explanation after the attack aligns with the wrong prediction, as if they were correct. The behavior occurs despite unchanged context, stem, and key ($\mathcal{C}$, $\mathcal{S}$, and $k$), which should contain the most important information for understanding and answering the question (Chai & Jin, 2004).

## 6 Discussion and future work

The present study illustrates that TCES is a compact yet effective way to construct adversarial distractors, which leads to significant performance degradation in LLMs. Our results suggest that to minimize model queries, using a budget of up to the inflection point on the query scaling curve is the most cost-effective. The perturbed texts produced by TCES appear natural and are less likely to be detected using grammar check or topical filtering (Morris et al., 2020a). Therefore, the setting can simulate realistic scenarios where the "attacks" may be initiated by unsuspecting users, such as healthcare professionals using an LLM-based clinical decision support system (Liu et al., 2023). In the examples illustrated in Fig. 5 and Appendix E, adversarial distractors can lead to disease misdiagnosis or misprescription of medication, which are detrimental scenarios that may occur in algorithmic decision-making in biomedicine using LLMs. Further improvement on the sampling attack performance may be achieved using learned samplers through rejection mechanisms (Narasimhan et al., 2024) or by finetuning the text embedding to improve the query efficiency of sampling. Alternatively, the semantic distance used for PDWS may be replaced with a concept distance (Choi et al., 2016) to improve attack performance on general text embeddings. Our approach may be integrated into interactive platforms for adversarial data collection or online monitoring systems for open-ended human-machine conversation (Chao et al., 2023).

Future research may focus on how to make LLMs more reliable through prompt engineering (Si et al., 2023; Nori et al., 2023) or leveraging retrieval from nonparametric knowledge bases (Weikum et al., 2021; Soman et al., 2023) or text sources (Wang et al., 2023b) to improve model generalization to long-tail entities. Besides vulnerability identification, the current work motivates adversarial defense strategies based on generalized distances (La Malfa et al., 2020) to establish a tiered system for robustness assessment in user-facing and domain-oriented applications to mitigate catastrophic failures.

## Acknowledgments

R.P.X. thanks N. Rethmeier for helpful technical discussions and we thank M. Collard for managing the computing resources. A.J.L. acknowledges funding from NIH T32GM067547, the UCSF Discovery Fellows Program, and the Achievement Rewards for College Scientists (ARCS) Fellowship. R.A.-A. would like to acknowledge funding from Weill Neurohub.

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
