# A  Adversarial distractor generation (general case)

We formulate here the general case of adversarial distractor generation when the choices contain text other than entities and one or more entities are present in a single choice. To keep the symbols consistent with Section 3, we let the question be $\mathcal{Q} = (\mathcal{C}, \mathcal{S}, \mathcal{D}, \mathcal{K})$, with the context $\mathcal{C}$, question stem $\mathcal{S}$. We write the distractor set $\mathcal{D} = \{\mathcal{D}_i\}_{i=1}^l$, and each distractor $\mathcal{D}_i$ contains a set of entities $\{d_{i,j}\}_{j=1}^{l_i}$ and additional text, which means that

$$\mathcal{D}_i \setminus (\cup_j \{d_{i,j}\}) \neq \varnothing. \tag{7}$$

The key $\mathcal{K}$ contains the entity $k$ and other text. We use the notation $\mathcal{Q}(d_{i,j} \rightarrow d'_{i,j})$ to denote that a distractor entity $d_{i,j}$ is substituted by $d'_{i,j}$ in the questions $\mathcal{Q}$. If $d_{i,j}$ in the question is adversarially perturbed, the model elicits a false response, while it answers correctly to the unperturbed question $\mathcal{Q}$,

$$\operatorname{argmax} \Pr_{\text{LLM}}(Y = \mathcal{K}|\mathcal{Q}(d_{i,j} \rightarrow d'_{i,j})) \neq \operatorname{argmax} \Pr_{\text{LLM}}(Y = \mathcal{K}|\mathcal{Q}). \tag{8}$$

The general case is reduced to the special case described in Section 3 when $l_i = 1$, $\mathcal{D}_i = d_{i,1} \coloneqq d_i$, and $\mathcal{K} = k$, which recover $\mathcal{D} = \{d_i\}_{i=1}^l$, and $\mathcal{Q} = (\mathcal{C}, \mathcal{S}, \mathcal{D}, k)$.

# B  Proof of the sampling regimes

Here we present a simple proof of the deterministic limit of the powerscaled distance-weighted sampling described in Eq. 2 of the main text. Although the distance function $h$ is not given a specific form, we assume it to satisfy the basic properties of a metric. Given a perturbation set $\mathcal{E}_{-k} = \{e_1, ..., e_m\}$ with respect to a key $k$, there exists a corresponding set of distance values $\mathcal{H}_{-k}$.

## B.1  The limit of $n \rightarrow +\infty$

We order the elements of $\mathcal{H}_{-k}$ and define the ordered set $\mathcal{H}_{-k,\uparrow} = \{h_{1,\uparrow}, ..., h_{m,\uparrow}\}$ such that,

$$h_{i,\uparrow} < h_{j,\uparrow} \quad \text{iff} \quad i < j \tag{9}$$

This indicates that $h_{m,\uparrow} = \max \mathcal{H}_{-k,\uparrow}$, and $h_{j,\uparrow}/h_{m,\uparrow} < 1$ for any $j \neq m$. We compute the probability weights as $n \rightarrow +\infty$ for each entity $e_j \in \mathcal{E}_{-k}$. When $j \neq m$,

$$\begin{aligned}
\lim_{n \rightarrow +\infty} p_{j,\uparrow} &= \lim_{n \rightarrow +\infty} \frac{h_{j,\uparrow}^n}{\sum_j h_{j,\uparrow}^n} \\
&= \lim_{n \rightarrow +\infty} \frac{(h_{j,\uparrow}/h_{m,\uparrow})^n}{\sum_{j \neq m} (h_{j,\uparrow}/h_{m,\uparrow})^n + (h_{m,\uparrow}/h_{m,\uparrow})^n} \\
&= \frac{0}{0+1} = 0.
\end{aligned} \tag{10}$$

Similarly, as $n \rightarrow +\infty$, the probability weight for the $m$th entity is

$$\lim_{n \rightarrow +\infty} p_{m,\uparrow} = 1. \tag{11}$$

For the entities in the perturbation set $\mathcal{E}_{-k}$, this indicates that only

$$e_j = \operatorname{argmax} \mathcal{H}_{-k,\uparrow} \equiv \operatorname*{argmax}_{e_j \in \mathcal{E}_{-k}} h(k, e_j) \tag{12}$$

is selected with certainty at the deterministic limit. The equivalence ($\equiv$) is used because the two sets $\mathcal{H}_{-k,\uparrow}$ and $h(k, e_j)$ ($j = 1, ..., m$) have the same elements. Eq. 12 also defines the farthest element sampling.

---

**Prompt format**

Answer the following question without explanation.

[Content]: A 23-year-old pregnant woman at 22 weeks gestation presents with burning upon urination. She states it started 1 day ago and has been worsening despite drinking more water and taking cranberry extract. She otherwise feels well and is followed by a doctor for her pregnancy. Her temperature is 97.7°F (36.5°C), blood pressure is 122/77 mmHg, pulse is 80/min, respirations are 19/min, and oxygen saturation is 98% on room air. Physical exam is notable for an absence of costovertebral angle tenderness and a gravid uterus.

[Question]: Which of the following are the best treatment for this patient?
A: Ampicillin
B: Ceftriaxone
C: Doxycycline
D: Nitrofurantoin

[Answer]:

---

### B.2 The limit of $n \to -\infty$

A similar approach to B.1 can be used to construct the probability weights in the limit $n \to -\infty$. As noted in the footnote of the main text, this limit is not attained since $k \notin \mathcal{E}_{-k}$. Nevertheless, we can define the deterministic sampling of the nearest entity to $k$ in a similar vein as Eq. 12,

$$e_j = \operatorname{argmin} \mathcal{H}_{-k,\uparrow} \equiv \underset{e_j \in \mathcal{E}_{-k}}{\operatorname{argmin}} h(k, e_j), \tag{13}$$

which we refer to as nearest element sampling.

### B.3 The case of $n = 0$

When $n = 0$, it is straightforward to show that $p_j|_{n=0} = 1/m$, regardless of $j$ using the definition in Eq. 2. This scenario therefore corresponds to uniform random sampling.

### B.4 Query-limited deterministic sampling

Given a query budget $B$, construct the ordered sets $\mathcal{H}_{-k,\uparrow}^{j,m} = \{h_{j,\uparrow}, ..., h_{m,\uparrow}\}$ ($j = 1, ..., B$), and assuming $B \leq m$, we know that $\mathcal{H}_{-k,\uparrow}^{j,m} \subseteq \mathcal{H}_{-k,\uparrow} \equiv \mathcal{H}_{-k,\uparrow}^{1,m}$ from the definition of $\mathcal{H}_{-k,\uparrow}$ in Appendix B.1. We use $B$-nearest element sampling to refer to the deterministic sampling carried out sequentially where at the $j$th ($j = 1, ..., B$) query, the sampled entity $e_j = \operatorname{argmin} \mathcal{H}_{-k,\uparrow}^{j,m}$. Similarly, we can construct the ordered sets $\mathcal{H}_{-k,\uparrow}^{1,m-j+1} = \{h_{1,\uparrow}, ..., h_{m-j+1,\uparrow}\}$ ($j = 1, ..., B$). We use $B$-farthest element sampling to refer to the sequential deterministic sampling where at the $j$th ($j = 1, ..., B$) query, the sampled entity $e_j = \operatorname{argmax} \mathcal{H}_{-k,\uparrow}^{1,m-j+1}$. When $B = 1$, the query-limited versions reduces to the farthest and nearest element sampling described earlier in Appendix B with only a single query.

## C Biomedical entity processing

The original `MedQA-USMLE` (Jin et al., 2021) and `MedMCQA` (Pal et al., 2022) datasets lack entity annotations. In the present work, we carried out a two-step annotation for both datasets.

**UMLS entity types** We annotated the datasets using the type unique identifiers (TUIs) specified in the UMLS (Bodenreider, 2004) for biomedical entities. The annotation produces TUIs and the entity boundaries for entire questions and was carried out using `scispaCy` (Neumann et al., 2019) with the en_core_sci_md model. The annotation should be regarded as algorithmically produced, rather than human-annotated and proofread ground truth.

**Binary mention labels**   For entity-level perturbation, the distinction between questions with and disease entity mentions is based on the type of entities present in the distractors. We generate the binary mention labels ("Drugs?" and "Diseases?") for each question selected from the two datasets using the UMLS semantic groups of the TUIs in (i) for the distractors. Specifically, a drug-mention question has one or more entities in the distractors of the *Chemicals & Drugs* semantic group. A disease-mention question has one or more entities of the *Disorders* semantic group in its distractors.

## D   Prompt formatting

We structured the prompt with bracketed tags to divide different parts (`[Context]`, `[Question]`, `[Answer]`). An example taken from the MedQA-USMLE dataset (Jin et al., 2021) is shown below. For the MedMCQA dataset (Pal et al., 2022), only the `[Context]` and `[Answer]`[1] tags were used because the questions are short and generally contain no context. The prompt formatting improves model performance and facilitates answer parsing in the model-generated text. The increased space between paragraphs in the following is only to assist viewing and is not included in the actual prompt.

## E   Model evaluation and attack examples

**Model inference runtime**   The evaluations were carried out on a variety of NVIDIA GPUs depending on the model size. For smaller models such as the Flan-T5 series (Chung et al., 2024), the runtime on each type of question (involving drug or disease mentions) in a dataset was generally between 10 and 30 minutes, while it was from one to more than four hours for larger model such as Flan-UL2 (Tay et al., 2022) and Palmyra-Med-20B (Kamble & Alshikh, 2023). The runtime for each type of questions is between two and four hours.

**Multi-point gradient estimation**   We demonstrate the effects of the quality of gradient estimation on the performance of the blackbox gradient attack (DiscreteZOO (Berger et al., 2021)) using a representative dataset in Fig. 6 using MedAlpaca-7B (Han et al., 2023) and the MedMCQA dataset (Pal et al., 2022). The evaluation used TCES introduced in Algorithm 1. According to Eq. 5, for each entity to attack, we randomly sampled two, five, and ten candidate points in its neighborhood in the embedding space for gradient estimation. The attack on each input instance proceeds with a fixed total of LLM queries (total query budget) as indicated on the horizontal axis. The results show that the larger the number of candidate points used for gradient estimation, the more effective the attack becomes before it consumes the total query budget, as measured by the attack success rate (ASR). Having too few candidate points can result in an early plateau phase in the ASR, which stunts its growth. Yet having more candidate points can slightly reduce the ASR growth in the rapid rise

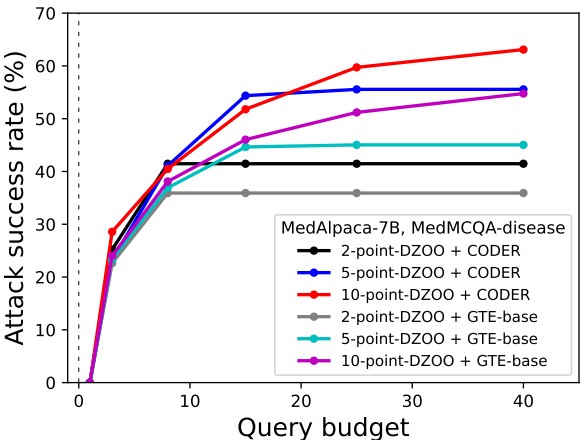

Figure 6: Quality of blackbox gradient estimation affects the scaling behavior of DiscreteZOO (DZOO) attacks using the CODER and GTE-base text embeddings.

phase of the scaling curve. The general trends are consistent in both the CODER (Yuan et al., 2022) and GTE-base (Li et al., 2023b) embeddings used for the illustration.

**Perturbed explanation in adversarial examples**   We list some examples of successful entity substitution attacks using adversarial distractors following Fig. 5 in the main text. For each example, the text box on top (the prompt formatting removed) shows the original choices on the left side and the perturbed choices

---

[1]Using the `[Context]` tag improves model performance than the `[Question]` tag here.

on the right side. The entity of interest before and after perturbation is colored in red and blue, respectively. The checkmark indicates the model prediction on unperturbed and perturbed questions. Beneath the text box are the corresponding heatmaps of token-wise Shapley values (or Shapley profile) before and after the entity substitution attack. The Shapley profiles for the MedQA-USMLE dataset (Jin et al., 2021) were generated using the model Flan-T5-large and those for the MedMCQA (Pal et al., 2022) using the model Flan-T5-xxl (Chung et al., 2024). The Shapley values were calculated using the partition explainer in `shap`, which uses the Owen values as an approximation (Chen et al., 2023).

---

**MedQA-USMLE**

An 18-year-old man presents with a sudden loss of consciousness while playing college football. There was no history of a concussion. Echocardiography shows left ventricular hypertrophy and increased thickness of the interventricular septum. Which is the most likely pathology underlying the present condition?

| | |
|---|---|
| A: Mutation in the myosin heavy chain ✓ | A: Mutation in the myosin heavy chain |
| B: Drug abuse | B: Drug abuse |
| C: Viral infection | C: Muller barth menger syndrome ✓ |
| D: Autoimmunity of myocardial fibers | D: Autoimmunity of myocardial fibers |

---

[Context]: An 18-year-old man presents with a sudden loss of consciousness while playing college football. There was no history of a concussion. Echocardiography shows left ventricular hypertrophy and increased thickness of the interventricular septum. [Question]: Which is the most likely pathology underlying the present condition?
A: Mutation in the myosin heavy chain
B: Drug abuse
C: Viral infection
D: Autoimmunity of myocardial fibers

[Context]: An 18-year-old man presents with a sudden loss of consciousness while playing college football. There was no history of a concussion. Echocardiography shows left ventricular hypertrophy and increased thickness of the interventricular septum. [Question]: Which is the most likely pathology underlying the present condition?
A: Mutation in the myosin heavy chain
B: Drug abuse
C: Muller barth menger syndrome
D: Autoimmunity of myocardial fibers

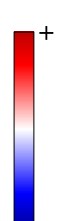

Unperturbed

Adversarially perturbed

**MedQA-USMLE**

A 41-year-old woman comes to the physician because of a 3-month history of anxiety, difficulty falling asleep, heat intolerance, and a 6-kg (13.2-lb) weight loss. The patient's nephew, who is studying medicine, mentioned that her symptoms might be caused by a condition that is due to somatic activating mutations of the genes for the TSH receptor. Examination shows warm, moist skin and a 2-cm, nontender, subcutaneous mass on the anterior neck. Which of the following additional findings should most raise concern for a different underlying etiology of her symptoms?

| | |
|---|---|
| A: Nonpitting edema ✓ | A: Nonpitting edema |
| B: Atrial fibrillation | B: Atrial fibrillation |
| C: Lid lag | C: Lid lag |
| D: Fine tremor | D: Nonpuerperal galactorrhea ✓ |

[Context]: A 41-year-old woman comes to the physician because of a 3-month history of anxiety, difficulty falling asleep, heat intolerance, and a 6-kg (13.2-lb) weight loss. The patient's nephew, who is studying medicine, mentioned that her symptoms might be caused by a condition that is due to somatic activating mutations of the genes for the TSH receptor. Examination shows warm, moist skin and a 2-cm, nontender, subcutaneous mass on the anterior neck. [Question]: Which of the following additional findings should most raise concern for a different underlying etiology of her symptoms?
A: Nonpitting edema
B: Atrial fibrillation
C: Lid lag
D: Fine tremor

Unperturbed

[Context]: A 41-year-old woman comes to the physician because of a 3-month history of anxiety, difficulty falling asleep, heat intolerance, and a 6-kg (13.2-lb) weight loss. The patient's nephew, who is studying medicine, mentioned that her symptoms might be caused by a condition that is due to somatic activating mutations of the genes for the TSH receptor. Examination shows warm, moist skin and a 2-cm, nontender, subcutaneous mass on the anterior neck. [Question]: Which of the following additional findings should most raise concern for a different underlying etiology of her symptoms?
A: Nonpitting edema
B: Atrial fibrillation
C: Lid lag
D: Nonpuerperal galactorrhea

Adversarially perturbed

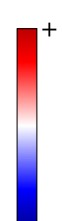

**MedQA-USMLE**

A 25-year-old man is brought to the emergency department after his girlfriend discovered him at home in a minimally responsive state. He has a history of drinking alcohol excessively and using illicit drugs. On arrival, he does not respond to commands but withdraws all extremities to pain. His pulse is 90/min, respirations are 8/min, and blood pressure is 130/90 mm Hg. Pulse oximetry while receiving bag-valve-mask ventilation shows an oxygen saturation of 95%. Examination shows cool, dry skin, with scattered track marks on his arms and legs. The pupils are pinpoint and react sluggishly to light. His serum blood glucose level is 80 mg/dL. The most appropriate next step in management is intravenous administration of which of the following?

| | |
|---|---|
| A: Naloxone | A: Naloxone |
| B: Phentolamine ✓ | B: Phentolamine |
| C: Methadone | C: Methadone ✓ |
| D: Naltrexone | D: Tienilic acid |

[Context]: A 25-year-old man is brought to the emergency department after his girlfriend discovered him at home in a minimally responsive state. He has a history of drinking alcohol excessively and using illicit drugs. On arrival, he does not respond to commands but withdraws all extremities to pain. His pulse is 90/min, respirations are 8/min, and blood pressure is 130/90 mm Hg. Pulse oximetry while receiving bag-valve-mask ventilation shows an oxygen saturation of 95%. Examination shows cool, dry skin, with scattered track marks on his arms and legs. The pupils are pinpoint and react sluggishly to light. His serum blood glucose level is 80 mg/dL.
[Question]: The most appropriate next step in management is intravenous administration of which of the following?
A: Naloxone
B: Phentolamine
C: Methadone
D: Naltrexone

Unperturbed

[Context]: A 25-year-old man is brought to the emergency department after his girlfriend discovered him at home in a minimally responsive state. He has a history of drinking alcohol excessively and using illicit drugs. On arrival, he does not respond to commands but withdraws all extremities to pain. His pulse is 90/min, respirations are 8/min, and blood pressure is 130/90 mm Hg. Pulse oximetry while receiving bag-valve-mask ventilation shows an oxygen saturation of 95%. Examination shows cool, dry skin, with scattered track marks on his arms and legs. The pupils are pinpoint and react sluggishly to light. His serum blood glucose level is 80 mg/dL.
[Question]: The most appropriate next step in management is intravenous administration of which of the following?
A: Naloxone
B: Phentolamine
C: Methadone
D: Tienilic acid

Adversarially perturbed

---

**MedQA-USMLE**

A 70-year-old woman presents to the office for a yearly physical. She states she has recently started experiencing pain in her legs and her back. Last year, she experienced a fracture of her left arm while trying to lift groceries. The patient states that she does not consume any dairy and does not go outside often because of the pain in her legs and back. Of note, she takes carbamazepine for seizures. On exam, her vitals are within normal limits. You suspect the patient might have osteomalacia. Testing for which of the following is the next best step to confirm your suspicion?

A: 25-hydroxyvitamin D ✓                    A: 25-hydroxyvitamin D
B: 1,25-hydroxyvitamin D                     B: 1,25-hydroxyvitamin D
C: Pre-vitamin D3                            C: Pre-vitamin D3 ✓
D: Dietary vitamin D2                        D: Stiripentol

---

[Context]: A 70-year-old woman presents to the office for a yearly physical. She states she has recently started experiencing pain in her legs and her back. Last year, she experienced a fracture of her left arm while trying to lift groceries. The patient states that she does not consume any dairy and does not go outside often because of the pain in her legs and back. Of note, she takes carbamazepine for seizures. On exam, her vitals are within normal limits. You suspect the patient might have osteomalacia.
[Question]: Testing for which of the following is the next best step to confirm your suspicion?
A: 25-hydroxyvitamin D
B: 1,25-hydroxyvitamin D
C: Pre-vitamin D3
D: Dietary vitamin D2

Unperturbed

[Context]: A 70-year-old woman presents to the office for a yearly physical. She states she has recently started experiencing pain in her legs and her back. Last year, she experienced a fracture of her left arm while trying to lift groceries. The patient states that she does not consume any dairy and does not go outside often because of the pain in her legs and back. Of note, she takes carbamazepine for seizures. On exam, her vitals are within normal limits. You suspect the patient might have osteomalacia.
[Question]: Testing for which of the following is the next best step to confirm your suspicion?
A: 25-hydroxyvitamin D
B: 1,25-hydroxyvitamin D
C: Pre-vitamin D3
D: Stiripentol

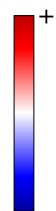

Adversarially perturbed

---

**MedMCQA**

Which of the following is granted orphan drug status for the treatment of Dravet's syndrome?

| | |
|---|---|
| A: Stiripentol | A: Stiripentol |
| B: Icatibant | B: Tocilizumab  ✓ |
| C: Pre-vitamin D3 | C: Pitolisant |
| D: Tafamidis  ✓ | D: Tafamidis |

---

[Context]: Which of the following is granted orphan drug status for the treatment of Dravet's syndrome?
A: Stiripentol
B: Icatibant
C: Pitolisant
D: Tafamidis

Unperturbed

[Context]: Which of the following is granted orphan drug status for the treatment of Dravet's syndrome?
A: Stiripentol
B: Tocilizumab
C: Pitolisant
D: Tafamidis

Adversarially perturbed

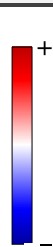

---

**MedMCQA**

Which among the following may be used as a sedatives-hypnotic?

| | |
|---|---|
| A: Zolmitriptan | A: Zolmitriptan  ✓ |
| B: Zileuton | B: Zileuton |
| C: Zolpidem  ✓ | C: N-acetylglucosamine |
| D: Zalcitabine | D: Zalcitabine |

---

[Context]: Which among the following may be used as a sedatives-hypnotic?
A: Zolmitriptan
B: Zileuton
C: Zolpidem
D: Zalcitabine

Unperturbed

[Context]: Which among the following may be used as a sedatives-hypnotic?
A: Zolmitriptan
B: Zileuton
C: N-acetylglucosamine
D: Zalcitabine

Adversarially perturbed