# OpenReview forum: "Assessing biomedical knowledge robustness in large language models by query-efficient sampling attacks"
_TMLR — Accepted by TMLR_

### Review · Reviewer_D6wa · 2024-10-10

**Summary Of Contributions:**

The paper investigates the adversarial robustness of LLMs in the biomedical domain to entity substitution.
- For this purpose, they adapt distance weight sampling to a sampling-based attack and additionally investigate an attack using zeroth order optimization.
- They demonstrate that currently available LLMs are vulnerable to adversarial entity substitutions and show that both attack types achieve higher attack success rates than random sampling (while still using a small query budget).
- They investigate connections between the adversarial vulnerability of the models and the model size and the effect of adversarial entities on model explainability

**Audience:**

Yes

**Broader Impact Concerns:**

I do not have broader impact concerns. The presented attack methods are valuable in studying the robustness of LLMs. However, I do not see a practical risk of such attacks in deployment scenarios.

**Claims And Evidence:**

Yes

**Requested Changes:**

- Could the authors provide one concrete example for the different variables in equation 3.1?
- I did not find a formal definition of $e$. In general, the paper is missing formal definitions of variables in some places (including their dimensionality, etc.).
- What is the original text span - $e_0$ can be understood as the embedding of the original text?
- Is eq (5) normally used in an iterative setting? Maybe indicate in the equation
- Would the authors consider open-sourcing an adversarial version of the base dataset as a “hard”
benchmark for biomedical Q&A (similar to ImageNet-A [1]). I believe this would be an interesting benchmark to measure the robustness of LLMs in pseudo-adversarial settings.

[1] Hendrycks et al., "Natural Adversarial Examples" 2021

**Strengths And Weaknesses:**

**Strengths:**

- Hyperparameters of the presented methods are evaluated extensively
- The paper makes precise claims with suitable experiments to back them up
- The authors introduce practical attacks with a reasonable query budget
- The method presented by the authors is suitable for generating difficult benchmark datasets in the biomedical Q&A domain, enabling researchers to investigate robust prediction methods

**Weaknesses:*

- The authors do not assess if recently proposed defenses to adversarial attacks in the LLM domain would improve the robustness of these models (However, I acknowledge that this might be out-of-scope of this work and do not expect the authors to provide such an experiment)
- A white-box evaluation would have been interesting to provide some lower bound on the robustness of the models used in the study. I would expect 100% ASR to be possible based on results in the literature in other domains.
-  The authors provide a method to investigate the robustness of LLMs to entity substitution. I believe the research community would have benefited from a "fixed dataset" with precomputed attacks as a sort of "difficult" benchmark in this domain.

---

> ### Author Response · Authors · 2024-10-23
> **Answer to reviewer D6wa**
>
> We thank the reviewer for their appraisal of our work. We have re-organized section 3.1 and 3.2 to emphasis the core definitions in the revision. The following are the answers to the requested changes.
>
> **Requested Changes:**
> > * Could the authors provide one concrete example for the different variables in equation 3.1?
>
> We use the example in Fig. 1b and extend the brief description in footnote 3. The stem $\mathcal{S}$ is the last sentence on the left side, "Which of the following is most likely to be elevated in this patient?'' The context $\mathcal{C}$ is all the text before the stem, "A 50-year-old woman ... Her pupils are dilated.'' The original distractor set $\mathcal{D}$ = \{Serum creatinine, Temperature, Creatine phosphokinase, Blood pressure\}, the perturbed distractor set $\mathcal{D}'$ = \{Serum creatinine, Temperature, Parathyroid hormone, Blood pressure\}, the key $k$ is Blood pressure, or equivalently, D
>
> We have updated our manuscript with the above as Example 3.1.
>
> > * I did not find a formal definition of $e$. In general, the paper is missing formal definitions of variables in some places (including their dimensionality, etc.).
>
> The symbol $e$ refers to the token sequence representation of a span.
>
> We have updated our manuscript with Definition 3.3: Given the token sequence representation of a span $e$, we define its corresponding vector representation in a text embedding of dimension $l$ as **e** $\in \mathbb{R}^l$.
>
> This definition also applies to other pairs of non-bolded and bolded letters used to represent spans such as ($k$, **k**), ($e'$, **e$'$**), ($e_0$, **e$_0$**), which we added notes to for clarification. If there are other symbols whose dimensionality needs clarification, we are happy to include in our response.
>
> > * What is the original text span $e_0$ - can be understood as the embedding of the original text?
>
> The symbol $e_0$ refers to the token sequence representation of the text span, the bold version **e$_0$** refers to the vector representation of the same text span. This is clarified in the previous question and Definition 3.3. In the example of Fig. 1b, $e_0$ refers to *Creatine phosphokinase*.
>
> > * Is eq (5) normally used in an iterative setting? Maybe indicate in the equation
>
> To our knowledge, Eq (5) in the implementation of DiscreteZOO was evaluated once for each ranked word to attack, because it allowed attacking multiple words sequentially. In our case, we disabled word ranking to recover the low-query-budget setting. We followed the exact implementation of Eq (5) in our benchmark.
>
> We also want to clarify that ZOO-based gradient estimators can be implemented in the iterative setting, but with a larger query budget, such as done in computer vision. Our goal here is to adhere to the original implementation of DiscreteZOO as much as possible while adapting it to the span-level entity substitution attack setting to compare with the sampling-based attacks.
>
> > * Would the authors consider open-sourcing an adversarial version of the base dataset as a “hard” benchmark for biomedical Q&A (similar to ImageNet-A [1]). I believe this would be an interesting benchmark to measure the robustness of LLMs in pseudo-adversarial settings.
>
> Yes, we will upload adversarial versions of the two biomedical QA datasets to a public repository.

---

> > ### Comment · Reviewer_D6wa · 2024-10-24
> > **Response**
> >
> > I thank the authors for their response. All my concerns were addressed. I reviewed the updated manuscript and found the respective sections to be considerably more precise. I recommend to accept the paper as it is.

---

### Review · Reviewer_HHFG · 2024-10-11

**Summary Of Contributions:**

This paper explores the robustness of biomedical knowledge in large language models (LLMs) by using a novel query-efficient sampling approach to generate adversarial entities. Specifically, it introduces type-consistent entity substitution (TCES) to collect adversarial examples, leveraging a gradient-free, embedding-space method called powerscaled distance-weighted sampling (PDWS). The paper makes a significant contribution by demonstrating the susceptibility of both pre-trained generalist and fine-tuned domain-specific LLMs to adversarial entity attacks in biomedical question-answering (QA) tasks. Moreover, the research reveals two regimes of adversarial entities (near and far in embedding space) and demonstrates how entity substitutions can effectively deceive token-wise Shapley value explanations.

**Audience:**

Yes

**Broader Impact Concerns:**

No concerns in this regard.

**Claims And Evidence:**

Yes

**Requested Changes:**

I would encourage the authors to address the “Weaknesses” listed above.

**Strengths And Weaknesses:**

**Strengths**:

1. TCES and PDWS for adversarial attacks on LLMs address the pressing need for model vulnerability assessment in specialized domains. TCES ensures entity consistency in biomedical contexts by maintaining semantic coherence, while PDWS efficiently explores embedding space to identify vulnerabilities with low query budgets.

2. The proposed gradient-free method for adversarial entity generation improves attack success rate with a low-query-budget setting. This efficiency is particularly relevant for practical robustness evaluations of LLMs.

3. The paper provides a rigorous experimental setup, evaluating multiple LLMs, such as Flan-UL2 and Palmyra-Med, across various biomedical datasets using metrics such as attack success rate (ASR), query efficiency, and model accuracy before and after adversarial perturbations.

4. The findings carry significant implications for the deployment of LLMs in high-stakes applications, such as healthcare, where model robustness and reliability are paramount.

*I am not familiar with entity substitution attacks in LLMs, but I hope to provide objective feedback on the paper's weaknesses and suggestions to improve the argumentation and presentation. If my suggestions are impractical or incorrect, I welcome discussion and corrections.*

**Weaknesses**:

1. This paper introduces the innovative PDWS, but the connection between DiscreteZOO in Sec. 3.3 and PDWS is not clear. Does the paper make any improvements to DiscreteZOO, or is it simply presented as an alternative and baseline to PDWS? Could DiscreteZOO be combined with PDWS to achieve better results?
2. The experiments only compare PDWS to random sampling. I suggest including two additional baselines related to PDWS: the 'farthest' and 'closest' methods used in [1], as they represent two extreme cases of $n$. This would effectively demonstrate the advantages of distance-based sampling.
3. I am curious why beam search and greedy search are not included as baseline methods for comparison. These methods are often known for high attack success rates, albeit at the cost of high query budgets.
4. One of the key advantages of the proposed TCES framework is that type-consistent substitutions make perturbed text harder to detect. While this point is stated in the paper, there is no empirical evidence to support it. I suggest providing verification experiments, such as calculating BERTScore[2] for tokens or questions before and after perturbation, to strengthen this argument.

**Reference**:

[1]: Berger, N., Riezler, S., Sokolov, A., & Ebert, S. (2021). Don’t Search for a Search Method — Simple Heuristics Suffice for Adversarial Text Attacks. ArXiv, abs/2109.07926.

[2]: Zhang, X., Zhao, J.J., & LeCun, Y. (2015). Character-level Convolutional Networks for Text Classification. Neural Information Processing Systems.

---

> ### Author Response · Authors · 2024-11-09
> **Reply to reviewer HHFG 1/2**
>
> We thank the reviewer for their appraisal of our work. We first address the weaknesses 1, 3, 4 in the following.
>
> **Weaknesses:**
> >1. This paper introduces the innovative PDWS, but the connection between DiscreteZOO in Sec. 3.3 and PDWS is not clear. Does the paper make any improvements to DiscreteZOO, or is it simply presented as an alternative and baseline to PDWS? Could DiscreteZOO be combined with PDWS to achieve better results?
>
> **Connection between PDWS and DiscreteZOO:** We considered a sampling view of adversarial attacks for text because of the discreteness and the finiteness of the viable search space (here referring to the perturbation set). Our descriptions of PDWS and DiscreteZOO in sections 3.2 and 3.3, especially Eqs. (3) and (6), unified the two approaches within the same distance-weighted sampling framework. Within the framework, we can distinguish the attacks by comparing the anchor point and the probability weights in the sampling procedure.
>
> **Improvements to DiscreteZOO:** We only intended to adapt DiscreteZOO to our attack setting to compare them on the same footing, because DiscreteZOO doesn’t immediately work with our attack setting. No explicit improvement is made to the gradient calculation step, which is the core of the DiscreteZOO algorithm. The key modifications are described in section 4.3 within the paragraph Attack Settings and Core Hyperparameters. To summarize, we adapted DiscreteZOO to (i) use the token boundary of the span for text substitution, (ii) enable split-model inference on multiple GPUs, (iii) make it compatible with decoder models. We also (iv) set the word ranking to be random (originally a setting that DiscreteZOO allows), (v) disabled the similarity threshold for candidate point selection. Step (i) is necessary because DiscreteZOO (and its dependency textattack) operates at the word level and (iv)-(v) have been used in the DiscreteZOO paper. The discovery we made along the way is that (iv) allows DiscreteZOO to run in the low-query-budget setting, which wasn't a primary concern for evaluating the adversarial robustness of BERT models.
>
> **Combining DiscreteZOO and PDWS:** DiscreteZOO can be combined with PDWS, indeed. We conducted preliminary studies during the completion of the current manuscript by comparing DiscreteZOO + RS (default setting, RS = random sampling) with DiscreteZOO + PDWS, where PDWS was used for selecting the candidate points for gradient estimation in Eq. (4). The intuition is that the gradient estimator will be more efficient with spatially biased sampling such as PDWS than RS. However, the difference is only considerable when $M$ > 10 in Eq. (4). We want to leave more detailed analysis and discussion on this topic for a future study.
>
> >3. I am curious why beam search and greedy search are not included as baseline methods for comparison. These methods are often known for high attack success rates, albeit at the cost of high query budgets.
>
> Beam search and greedy search are known to be less query-efficient than DiscreteZOO and sampling-based attacks, as the reviewer has remarked. They don't operate in the low-query-budget setting, which is a selection criterion of our benchmarks. Our focus here is on query efficiency and the diversity of (adversarial) samples. We think the focus solely on attack success rate in the large-query-budget setting in traditional text adversarial attack literature is insufficient and impractical in addressing the evaluation challenges for LLMs. In addition, classic beam and greedy search methods don't work in our setting since we only replace one entity to keep perturbations low.
>
> >4. One of the key advantages of the proposed TCES framework is that type-consistent substitutions make perturbed text harder to detect. While this point is stated in the paper, there is no empirical evidence to support it. I suggest providing verification experiments, such as calculating BERTScore[2] for tokens or questions before and after perturbation, to strengthen this argument.
>
> We thank the reviewer for the suggestion. However, the questions in the MedQA-USMLE dataset are generally long, so are beyond the token limit acceptable by BERTScore. We therefore opt to verify the semantic distortion before and after entity substitution using SentenceTransformer along with a pretrained RoBERTa-large model, which can compare the entire prompt used as input to the LLMs evaluated in this work. We calculated the semantic similarity between the prompts before and after substitution. The averaged prompt semantic similarity (PSS) are shown in Fig. 2e-h for each different attack conditions. The scores are generally high and show consistency across models. We discussed the semantic distortion evaluation in two paragraphs (labeled in blue) in section 5.1.
>
> For the record, we calculated the BERTScore for the MedMCQA dataset because the questions are shorter. The results have similar trends as the average PSS calculated using the SentenceTransformer.

---

> ### Author Response · Authors · 2024-11-13
> **Reply to reviewer HHFG 2/2**
>
> Sorry about the delay, we next address weakness 2 pointed out by the reviewer.
>
> >2. The experiments only compare PDWS to random sampling. I suggest including two additional baselines related to PDWS: the 'farthest' and 'closest' methods used in [1], as they represent two extreme cases of $n$. This would effectively demonstrate the advantages of distance-based sampling.
>
> >[1]: Berger, N., Riezler, S., Sokolov, A., & Ebert, S. (2021). Don’t Search for a Search Method — Simple Heuristics Suffice for Adversarial Text Attacks. ArXiv, abs/2109.07926.
>
> We thank the reviewer for this helpful suggestion. We implemented the two deterministic sampling methods that the reviewer has suggested along with their query-limited version, which only takes the $B$-nearest or the $B$-farthest samples ($B$ is the query budget). A formal definition is given in Appendix B.4. For convenience of reference, we call the single-query version of the two types of deterministic sampling the nearest element sampling and farthest element sampling, respectively. We experimented with deterministic sampling in the same text embeddings used for comparing PDWS and DiscreteZOO attacks in this work.
>
> We emphasize that firstly, the tunability of PDWS allows to sample different regions in the attack surface (i.e. entity substitution), which is not viable by only looking at the nearest and farthest elements within the perturbation set through deterministic sampling. Secondly, we studied the query scaling behavior of deterministically sampling the $B$ nearest and farthest samples ($B$-nearest and the $B$-farthest) for the two cases in Fig. 4 (Flan-T5-xl and MedAlpaca). The added Figs. 4c-d show that deterministic sampling has competitive advantages in both performance and scaling against DiscreteZOO and is not far from that of PDWS. Deterministic sampling is more competitive in a specialized text embedding (such as CODER) than a general one (such as GTE-base), although in both cases the PDWS attack still outperforms its deterministic counterparts. We discussed this point in the paragraph **Deterministic sampling attacks are competitive proxies of their probabilistic counterparts** before section 5.3.
>
> Beside the query scaling behavior of deterministic sampling, we also conducted some evaluation on the single-query attack success rates. PDWS largely outperforms its deterministic sampling counterparts, demonstrating the advantages of PDWS. We could provide this as a supplementary table per the reviewer's request.

---

> > ### Comment · Reviewer_HHFG · 2024-11-14
> >
> > Thank you to the authors for their thorough responses. All of my concerns have been addressed. I am pleased with the additional experiments on semantic perturbation and the farthest & nearest methods. Based on these improvements, I recommend accepting the paper as is.

---

### Review · Reviewer_rYjV · 2024-10-16

**Summary Of Contributions:**

This work propose a entity substitution attack to examine the vulnerabilities of biomedical LLMs.
It proposed an embedding-distance-weighted sampling (DWS) method to generate adversarial distractors, which is query efficiency and scalable.
Experiments demonstrate the effectiveness of DWS and the brittleness of biomedical knowledge of LLMs.

**Audience:**

Yes

**Claims And Evidence:**

Yes

**Requested Changes:**

- More powerful and decoder-only model should be evaluated, at least ChatGPT and a powerful BLM in the leaderboard.
- Add the investigation and explaination of cosine distance in the biomedical context.
- A justification or additional experiments about the validity of adversarial distractors.
- It's better to provide a defensive method against the attack.

**Strengths And Weaknesses:**

## Strength

- The research problem of evaluating the domain knowledge of large language models (LLMs) is highly significant.
- The proposed method is query efficiency and can be scaled to larger model and benchmark.
- The evaluation covers both sampling-based and search-based attack methods.


## Weakness

- The implementation of cosine distance in the NE model requires further discussion. What does cosine similarity of model embeddings represent in the biomedical context? Does a small distance reflect similar drug properties, or just token similarity (e.g., `Zolmitriptan` vs `Zolpidem`)? One way to investigate this is by evaluating the sentence similarity of drugs and disease descriptions.
- The validity of adversarial distractors should be examined. Based on the results in Table 1, the PWS approach perform better when n < 0 in most cases, suggesting that the adversarial distractors are very close to the true answers in embedding space. This raises the possibility that these distractors could also be true.  The authors should provide a justification for this and conduct additional experiments.
- Only few models are tested. The tested Flan series models are relatively weaker and do not follow the mainstream decoder-only architecture. Testing more powerful LLMs, such as LLaMA3, GPT or BLM in the leaderboard [1], would benefit the community and increase the impact of this work.
- The work would benefit from including a defensive method that could enhance the robustness of BLMs against entity substitution attacks.


## Ref
[1] https://huggingface.co/spaces/openlifescienceai/open_medical_llm_leaderboard

---

> ### Author Response · Authors · 2024-11-12
> **Response to reviewer rYjV 1/4**
>
> We thank the reviewer for their assessments on our work. We intend to break up the answers into a few different parts.
>
> >**Weakness [W] + Requested Changes [C]**
>
> >[W] The implementation of cosine distance in the NE model requires further discussion. What does cosine similarity of model embeddings represent in the biomedical context? Does a small distance reflect similar drug properties, or just token similarity (e.g., Zolmitriptan vs Zolpidem)? One way to investigate this is by evaluating the sentence similarity of drugs and disease descriptions.
>
> >[C] Add the investigation and explaination of cosine distance in the biomedical context.
>
> We believe this request has already been largely addressed in the CODER paper [1], where cosine distance was used in the contrastive learning framework. Specifically, cosine distance was discussed in Section 3.3 of [1], where the authors explained that it indicates the semantic similarity of the biomedical terms. We also want to remind the reviewer that cosine distance was also historically used for measuring the semantic similarity of biomedical terms, long before BERT-sized models came to existence [2].
>
> [1] Yuan et al., CODER: Knowledge-infused cross-lingual medical term embedding for term normalization, J. Biomed. Inform. (2022), arXiv:2011.02947
>
> [2] Garcia & Brandt, Semantic similarity in the biomedical domain: an evaluation across knowledge sources, BMC Bioinform. (2012).

---

> ### Author Response · Authors · 2024-11-12
> **Response to reviewer rYjV 2/4**
>
> >**Weakness [W] + Requested Changes [C]**
>
> >[W] The validity of adversarial distractors should be examined. Based on the results in Table 1, the PWS approach perform better when n < 0 in most cases, suggesting that the adversarial distractors are very close to the true answers in embedding space. This raises the possibility that these distractors could also be true. The authors should provide a justification for this and conduct additional experiments.
>
> >[C] A justification or additional experiments about the validity of adversarial distractors.
>
> We mentioned in the dataset section (section 4.1) that the entities in the perturbation set ($\mathcal{E}_{-k}$) are unique. The perturbation set is defined as $\mathcal{E}$ with the entity in the key $k$ removed. This makes sure that there is no entities identical to the key in the perturbation set, so the sampled distractor entities for substitution cannot be accidentally true.
>
> In the revision (section 5.1 and especially under **Adversarial entity characteristics**), we have given an example of adversarial disease names (*diabetes*) that are close to the key (*diabetes II*). Here, diabetes is an adversarial distractor when diabetes II is the key because the name diabetes would be insufficiently detailed for characterizing the disease diagnosis. The treatment plans for diabetes II and diabetes I, both being disease subtypes that fall under diabetes, are very different. Similar adversarial entities could arise when using incorrect disease subtypes, which is an important issue in real-world deployment of these models.

---

> ### Author Response · Authors · 2024-11-12
> **Response to reviewer rYjV 3/4**
>
> >**Weakness [W] + Requested Changes [C]**
>
> >[W] The work would benefit from including a defensive method that could enhance the robustness of BLMs against entity substitution attacks.
>
> >[C] It's better to provide a defensive method against the attack.
>
> We thank the reviewer for this request. Our main goal of this work is to highlight a vulnerability of great interest for the use of LLMs with biomedical knowledge. The work also examines, from the perspective of robustness evaluation, how to efficiently explore the attack surface (i.e. entity substitution). We believe the defense strategies for this type of attacks are worth investigation but they are out of scope for this work. As a start, adversarial training, or more practically, adversarial finetuning methods could be a viable defense, which will need the adversarial datasets generated from this work. In addition, a version of randomized smoothing compatible with the discrete text space, or simply averaging over answers from permuting the question choices may also reduce the attack success rate to some extent. For four-choice QA, there can be 24 permutations for each question, which may be used for smoothing. These latter two methods would increase the number of evaluations on LLMs significantly and there should be a tradeoff between the density (or spatial distribution) of adversarial entities in the embedding space and the efficiency of the defense. We think these questions are worth exploring dedicatedly in a separate work.
>
> We could offer some discussion on the potential defense methods in the revised Discussion (section 6) per the reviewer's request.

---

> ### Author Response · Authors · 2024-11-13
> **Response to reviewer rYjV 4/4**
>
> >**Weakness [W] + Requested Changes [C]**
>
> >[W] Only few models are tested. The tested Flan series models are relatively weaker and do not follow the mainstream decoder-only architecture. Testing more powerful LLMs, such as LLaMA3, GPT or BLM in the leaderboard [1], would benefit the community and increase the impact of this work.
>
> [1] https://huggingface.co/spaces/openlifescienceai/open_medical_llm_leaderboard
>
> >[C] More powerful and decoder-only model should be evaluated, at least ChatGPT and a powerful BLM in the leaderboard.
>
> We didn't orirginally benchmark on the very latest models largely because of their known limited inference speed (e.g. LLaMA3) and cost (e.g. OpenAI models). Given our budget and computing resource constraints, the models we chose for evaluations tend to have a balanced speed and knowledge to illustrate the point. The observed effects in this work seem to be more affected by the model size than its release date.
>
> **ChatGPT model:** We have included single-query evaluations and attacks on GPT-3.5 in Table 1 on the MedQA-USMLE dataset.
>
> **High-ranking BLM on the leaderboard:** We have added evaluations on the MedLlama-3-8B-v2.0 biomedical language model from John Snow Labs (https://huggingface.co/johnsnowlabs/JSL-MedLlama-3-8B-v2.0), which is the highest-ranking version (on Open Medical LLM Leaderboard) of the model from the same lab that we have access to. The zero-shot accuracy of this model on medical QA tasks is in the range 0.58 - 0.83, which is higher than all our previous models. Nevertheless, for the MedMCQA dataset that has been evaluated, the two-regime effect persists the PDWS attack yields higher attack success rates. We have included the results on MedMCQA dataset into Table 1.

---

> ### Comment · Reviewer_rYjV · 2024-11-17
>
> I thank the authors for their detailed responses. All of my concerns have been addressed. I am pleased with the additional experiments about mainstream LLMs and the clarification of cosine distance. Thus, I recommend accepting the paper as is.

---

### Decision · Action_Editor_RC1T · 2024-11-23

**Recommendation:** Accept as is

**Comment:**

The reviewers unanimously agree that the paper is ready to be accepted.

**Audience:**

Both LLMs and adversarial attacks are relevant to the ML community and as such this paper is indeed very relevant to the TMLR audience.

**Claims And Evidence:**

The paper investigates the adversarial robustness of large language models (LLMs) in the biomedical domain by introducing type-consistent entity substitution (TCES) and powerscaled distance-weighted sampling (PDWS). It demonstrates that domain-specific LLMs are vulnerable to these adversarial attacks, achieving higher success rates than random sampling while using a small query budget.

The reviewers agree that this is an interesting and relevant observation for the community.

---

> ### Author Response · Authors · 2024-12-12
> **Camera-ready version and response to the decision**
>
> We sincerely thank the editors and reviewers for their effort in reviewing the manuscript, which helped reshape and improve its content. In the camera-ready version, we have made the following adjustments.
>
> * Text adjustments of the abstract.
> * Performed typo fixes throughout the manuscript.
> * Changed the colored text during revision (blue) back to the normal color (black).
> * Shifted the location of the figures to align with their mentions in text.
> * Split up two long paragraphs in Sections 1 (last one) and 5.1 (first one).
> * Updated footnotes to include links to embedding models.
> * Added an acknowledgement section before references.
> * De-anonymized the author information and included a link to code.